# Exosomal circEZH2_005, an intestinal injury biomarker, alleviates intestinal ischemia/reperfusion injury by mediating Gprc5a signaling

Wenjuan Zhang[1,2], Bowei Zhou[1,2], Xiao Yang[1], Jin Zhao[1], Jingjuan Hu[1], Yuqi Ding[1], Shuteng Zhan[1], Yifeng Yang[1], Jun Chen[1], Fu Zhang[1], Bingcheng Zhao [1], Fan Deng[1], Zebin Lin [1], Qishun Sun[1], Fangling Zhang[1], Zhiwen Yao[1], Weifeng Liu[1], Cai Li [1] ✉ & Ke-Xuan Liu [1] ✉

Intestinal ischemia/reperfusion (I/R) injury is a severe clinical condition without optimal diagnostic markers nor clear molecular etiological insights. Plasma exosomal circular RNAs (circRNAs) are valuable biomarkers and therapeutic targets for various diseases, but their role in intestinal I/R injury remains unknown. Here we screen the expression profile of circRNAs in intestinal tissue exosomes collected from intestinal I/R mice and identify circEZH2_005 as a significantly downregulated exosomal circRNA. In parallel, circEZH2_005 is also reduced in the plasma of clinical cardiac surgery patients who developed postoperative intestinal I/R injury. Exosomal circEZH2_005 displays a significant diagnostic value for intestinal injury induced by I/R. Mechanistically, circEZH2_005 is highly expressed in intestinal crypt cells. CircEZH2_005 upregulation promotes the proliferation of Lgr5+ stem cells by direct interaction with hnRNPA1, and enhanced Gprc5a stability, thereby alleviating I/R-induced intestinal mucosal damage. Hence, exosomal circEZH2_005 may serve as a biomarker for intestinal I/R injury and targeting the circEZH2_005/hnRNPA1/Gprc5a axis may be a potential therapeutic strategy for intestinal I/R injury.

Intestinal ischemia/reperfusion (I/R) is a common and serious clinical problem that leads to intestinal injury and dysfunction, or even failure, of multiple organs[1]. Although it is associated with a high mortality and complication rate, acute intestinal ischemia can usually be reversed if diagnosed and treated in time[2,3]. To improve its early diagnosis, several studies have suggested measuring different biomarkers, including intestinal fatty acid binding protein (IFABP), diamine oxidase, glutathione S-transferase, citrulline, and smooth muscle protein[4–6]. Although most of these biomarkers have proven valuable in preclinical studies, their use in the clinical setting is limited by several shortcomings, including low sensitivity and specificity and poor reliability and accuracy[7,8]. Therefore, more sensitive and specific markers of intestinal injury are urgently needed.

To date, the underlying mechanism of intestinal I/R remains poorly understood. Intestinal injury caused by I/R involves all aspects of the intestinal mucosa's mechanical, immune, and microbial barriers[9,10]. However, the precise molecular factors involved in these processes need to be elucidated, as shown by the low survival rate of

[1]Department of Anesthesiology, Nanfang Hospital, Southern Medical University, Guangzhou, 1838 Guangzhou Avenue North, Guangzhou 510515, China. [2]These authors contributed equally: Wenjuan Zhang, Bowei Zhou. ✉e-mail: licaisysu@163.com; liukexuan705@163.com

patients with intestinal I/R[11,12]. Therefore, in-depth molecular investigations of intestinal I/R mechanisms are essential for the development of efficient treatment strategies.

circRNAs are a class of noncoding RNAs characterized by covalently closed loop structures with neither 5′ to 3′ polarity nor a polyadenylated tail[13]. circRNAs are highly abundant and are expressed in cell type-, tissue-, and stage-specific patterns[14]. In addition, circRNAs are implicated in a wide range of physiological and pathological processes, such as cell survival, growth, and differentiation[15]. Exosomes are nanoscale extracellular vesicles that are released from all cells in the body and are regarded as valuable circulating biomarkers for various diseases as they harbor abundant RNAs, proteins, and other biological components[16,17]. Importantly, circRNAs have been demonstrated to be stable and are highly enriched in exosomes[18]. They are highly stable compared with their parental linear RNAs and are easily measured in circulating exosomes because they lack exonuclease binding sites[19–21]. Therefore, exosomal circRNAs can be potential biomarkers of disease progression[22]. For example, exosomal circRNAs are valuable markers for effectively identifying the severity of alcohol dependence in patients with chronic alcohol consumption[23]. Exosomal circRNA-104484 and circRNA-104670 can be used as biomarkers for sepsis[24]. In addition to their use as biomarkers, exosomal circRNAs are involved in pathological processes in numerous diseases, including ischemic diseases. circHIPK3 is released in exosomes of hypoxia-pretreated cardiomyocytes and regulates the oxidative damage of cardiovascular endothelial cells through the miR-29a/IGF-1 pathway[25]. Neuron-derived exosomal circRNA has therapeutic potential against ischemic stroke[26]. However, whether exosomal circRNA expression is altered in the I/R-damaged intestinal mucosa and the potential therapeutic effect of targeting these circRNAs are currently unclear.

Based on previous reports, in this study, we aimed to discover a circulating exosomal circRNA with diagnostic potential in the setting of early intestinal I/R injury. We initially performed high-throughput sequencing of circRNA to characterize differential expression profiles of circRNA in intestinal tissue exosomes after I/R injury. The results were validated through intestinal I/R mice tissue, cell line, and exosome analyses. Subsequently, we evaluated the diagnostic power of target exosomal circRNAs in plasma obtained from patients with intestinal I/R injury after cardiopulmonary bypass (CPB) surgery. Furthermore, in vitro and in vivo experiments with cell and mouse models were conducted to comprehensively clarify the biological function of a target exosomal circRNA involved in intestinal I/R injury.

## Results

### Exosomal circEZH2_005 expression is associated with intestinal injury induced by I/R

To determine whether exosomal circRNA could be indicative of ischemia-induced intestinal injury, we isolated intestinal exosomes from mice with intestinal I/R (1 h of ischemia and 2 h of reperfusion) and sham-operated controls. The intestinal exosomes were characterized under a transmission electron microscope, nanoparticle tracking analysis, and western blotting (Fig. 1a–c). Subsequently, circRNA sequencing was performed to obtain the circRNA profile of these exosomes (Supplementary Fig. 1a). After establishing the filter criteria (fold change >2 or <0.5, and $P < 0.05$), 20 circRNAs were differentially and highly expressed in intestinal I/R exosomes (Fig. 1d–f). We identified the top five differentially expressed circRNAs for the putative target circRNA, among which, circ-Eya3 and circ-Lbr were upregulated and circEZH2_005, circ-Tmem267, and circ-Herc3 were downregulated, which corresponded with the sequencing results (Supplementary Fig. 1b, c). The presence of four differentially expressed circRNAs in the intestinal tissue of mice was

confirmed through Sanger sequencing, except for circ-Tmem267 (Supplementary Fig. 1d, e).

We analyzed the expression of the above four circRNAs in intestinal I/R mice at different time points of I/R (Supplementary Fig. 2a–c). Overall, circ-Eya3, circ-Lbr, circEZH2_005, and circ-Herc3 showed the same expression trend in mouse plasma exosomes as observed in intestinal tissue and intestinal exosomes, among which, circEZH2_005 was the most altered circRNA and was validated to be downregulated within the early hours of I/R injury (Supplementary Fig. 3a–c). These results were further confirmed in intestinal epithelial cells (Mode-K cells) subjected to different durations of H/R (Supplementary Fig. 4a, b). Furthermore, plasma exosomal circEZH2_005 expression was negatively correlated with Chiu's score and IFABP levels (Fig. 1g, h). This highlights the potential relationship between exosomal circEZH2_005 and intestinal I/R injury.

### Exosomal circEZH2_005 as a biomarker to distinguish intestinal I/R injury in patients

circEZH2_005 shares high homology (95% identity) between humans and mice and its downregulated expression in exosomes was related to intestinal injury induced by I/R. Therefore, we hypothesized that plasma exosomal circEZH2_005 might function as a specific biomarker for intestinal I/R injury diagnosis (Supplementary Fig. 5a). To test this hypothesis, we extracted plasma exosomes (Supplementary Fig. 5b, c) from 50 patients undergoing CPB surgery. During the perioperative period, the gut suffers from I/R and oxidative stress, which lead to enterocyte injury[27,28]. CPB has been used as a clinical model of intestinal I/R[29,30]. No significant differences in baseline characteristics were observed between the ischemic and non-ischemic groups except for age (Supplementary Table 1). Notably, qRT-PCR analysis showed that circEZH2_005 expression gradually decreased right after perfusion, peaked 6 h after perfusion in patients with AGI grades II, III, and IV, and remained stable until 12 h after perfusion (Fig. 2a). However, this dynamic was not found in AGI < II group patients without gastrointestinal injury (GI) complications (Supplementary Fig. 5d). Moreover, widely used circulating markers of enterocyte damage, IFABP, and D-lactate, were detected in all groups. The dynamic peak value changes in plasma circEZH2_005 showed a pattern of variation similar to that of IFABP (Fig. 2b, c and Supplementary Fig. 5e, f). Plasma exosomal circEZH2_005 was negatively correlated with IFABP and D-lactate levels (Fig. 2f, g).

In receiver operating characteristic analysis, circEZH2_005, as well as D-lactate and IFABP, exhibited favorable diagnostic potential, with area under the curve (AUC) values of 0.732 (95% confidence interval, 0.578–0.886), 0.685 (0.471–0.899), and 0.776 (0.625–0.928), respectively. When combing exosomal circEZH2_005 and IFABP in the logistic model, the panel displayed more robust diagnostic accuracy for intestinal I/R injury, with an AUC of 0.846 (0.700–0.993) (Fig. 2d, e). As expected, we observed a negative correlation between exosomal circEZH2_005 and AGI score (Fig. 2h). These data highlight the diagnostic potential of exosomal circEZH2_005 for the identification of patients with postoperative intestinal I/R injury.

### Characterization and expression of circEZH2_005

circEZH2_005 is generated from the circularization of exons 2 and 3 of the Ezh2 located on chromosome 9 (113,734,352–113,735,838), as evidenced by Sanger sequencing (Fig. 3a). In mice, it is highly expressed in the intestinal tract compared with other tissues (e.g., liver, kidney, brain, and lungs) (Fig. 3b). circEZH2_005 is a stable biomolecule as demonstrated by both the RNase R digestion experiment and its half-life (Fig. 3c, d). To visualize the cellular distribution and relative abundance of circEZH2_005 in intestinal tissues, we performed fluorescence in situ hybridization (FISH) assays and found that circEZH2_005 was preferentially localized in the jejunal crypts (Fig. 3e). In particular, its expression was

 

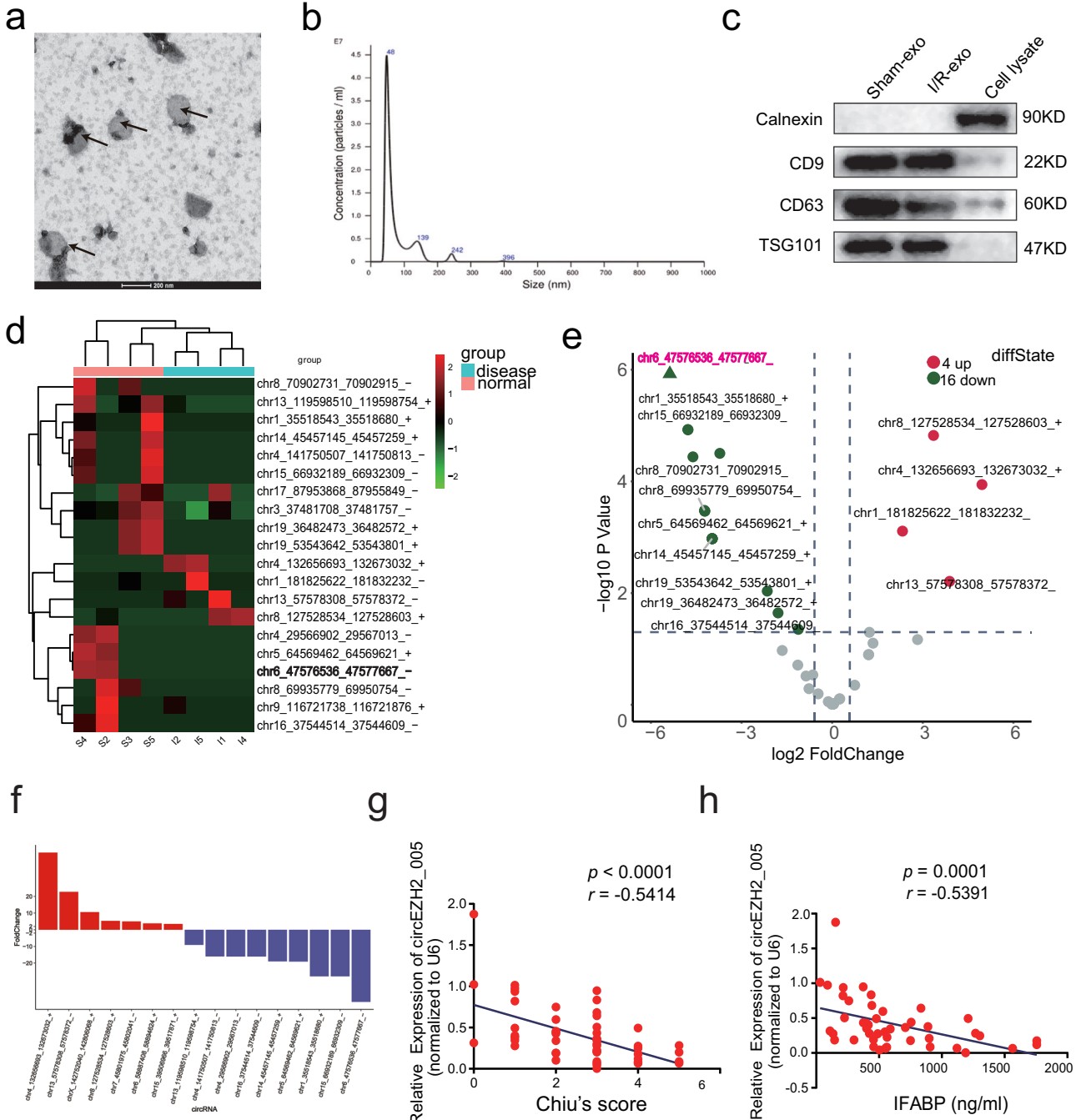

**Fig. 1 | Identification and screening of exosomal circRNAs that are associated with intestinal injury in intestinal I/R mode mice. a** Characterization of exosomes isolated from mice intestinal tissue, TEM was performed to confirm the shape of the exosomes. **b** The size of exosomes was measured by NTA. **c** The exosome-specific markers CD9, CD63, TSG101, and Calnexin were shown by western blot. **d** Heatmap showed twenty differentially expressed circRNAs in the ischemic intestinal tissues by circRNA-seq of intestinal tissue exosomes. **e** The volcano plot analyzed the significantly up-(red) and down-regulated (green) circRNAs based on fold-change (FC ≥ |2| and $p < 0.05$). The statistical model used for the volcano map data was negative binomial models, and the test method was generalized linear models, Benjamini & Hochberg for adjustments were made for multiple comparisons. **f** Differential expression analysis of circRNAs in results described in values is the means from 4 animals. **g** Correlation between circEZH2_005 and chiu's score was performed ($n = 54$ biological animals) using Spearman correlation analysis, $r = -0.5414$, $p < 0.0001$, 95% confidence interval (−0.7107 to −0.3127). **h** Correlation between circEZH2_005 and IFABP was performed ($n = 45$ biologically animals) using Spearman correlation analysis, $r = -0.5391$, $p = 0.0001$, 95% confidence interval (−0.7232 to −0.2836). Results are expressed as the means ± SD. (*$p < 0.05$, **$p < 0.01$, ***$p < 0.001$). Source data are provided as a Source data file.

markedly decreased after intestinal I/R treatment. In accordance with FISH data, qRT-PCR showed that the expression of circEZH2_005 in intestinal Lgr5+ crypt stem cells was higher than that in Lgr5- cells and was significantly reduced after intestinal I/R treatment (Fig. 3f). Moreover, circEZH2_005 was highly expressed in Lgr5+ intestinal stem cells (ISCs) (Fig. 3g). Analysis of nuclear and cytoplasmic RNA revealed that circEZH2_005 was predominantly expressed in the cytoplasm (Fig. 3h).

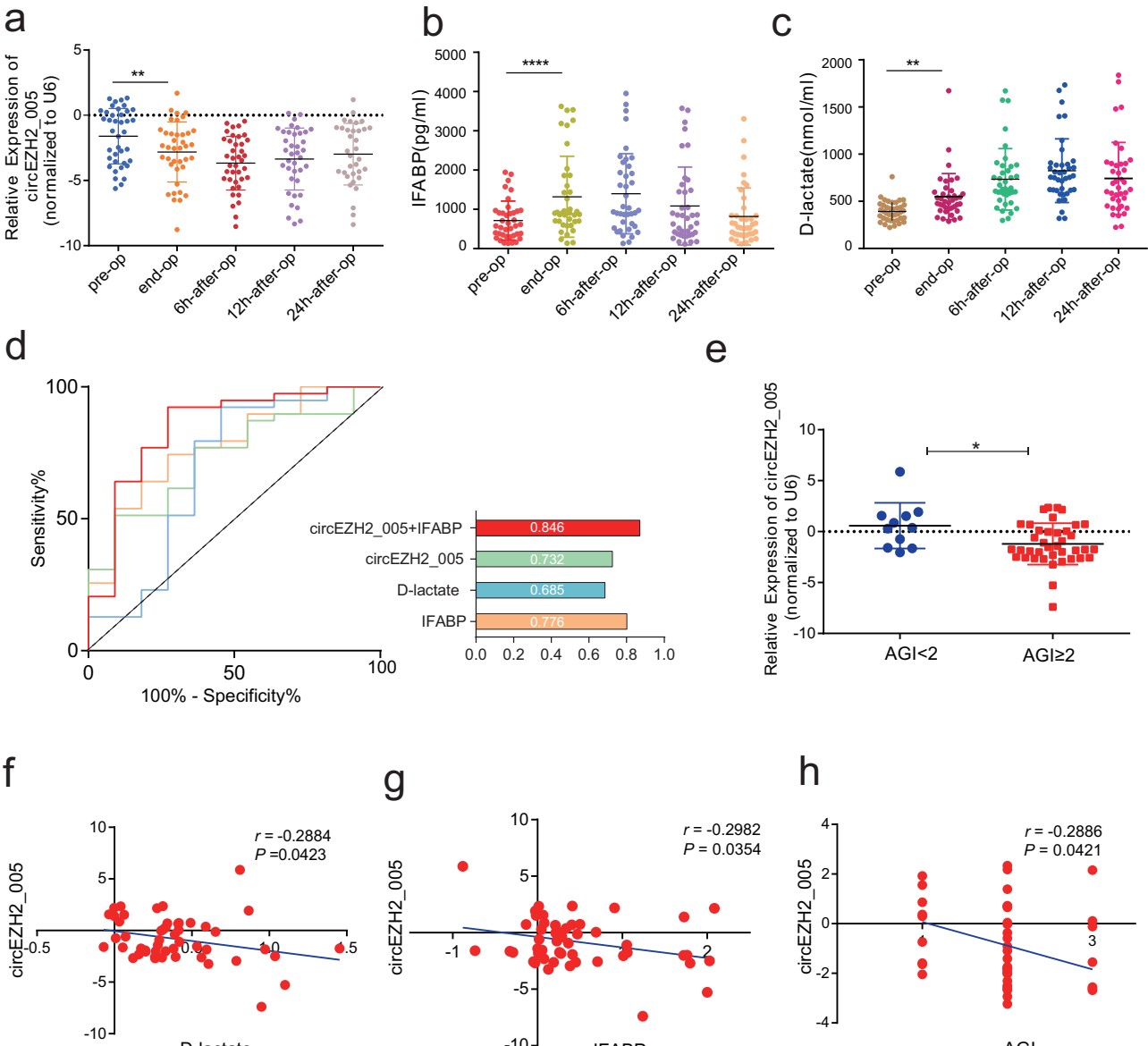

**Fig. 2 | Identification of exosomal circEZH2_005 as a biomarker for intestinal I/R injury in patients undergoing cardiopulmonary bypass surgery. a** RT-qPCR analysis of exosomal circEZH2_005 expression in the plasma of patients with intestinal I/R injury in preoperative, end operative, 6 h after operative, 12 h after operative, and 24 h after operative ($n = 39$ in the intestinal injury group). End-op vs pre-op, $p = 0.0093$. **b** Plasma I-FABP and D-lactate levels were assessed by ELISA ($n = 39$ in the intestinal injury group). End-op vs pre-op, $p < 0.0001$. **c** Plasma D-lactate levels were assessed by ELISA ($n = 39$ in the intestinal injury group). End-op vs pre-op, $p = 0.0040$. **d** ROC curves of I-FABP, D-lactate, and exosomal cir-cEZH2_005 alone or in combination for postoperative intestinal I/R injury ($n = 11$ in controls, $n = 39$ in the intestinal injury group). **e** RT-qPCR analysis of exosomal circEZH2_005 expression levels in the plasma of patients diagnosed with intestinal I/R injury ($n = 39$) and without intestinal I/R injury ($n = 11$). $p = 0.0152$. **f, g** The correlation analysis between the levels of exosomal circEZH2_005 and the levels of D-lactate was performed ($n = 50$) using Pearson correlation, $p = 0.0423$, $r = -0.2884$, 95% confidence interval ($-0.5246$ to $-0.01089$). The correlation analysis between the levels of exosomal circEZH2_005 and the levels of IFABP was performed ($n = 50$) using Pearson correlation, $p = 0.0354$, $r = -0.2982$, 95% confidence interval ($-0.5324$ to $-0.02167$). **h** The correlation analysis between the levels of exosomal circEZH2_005 and (AGI) grade was performed ($n = 50$) using Spearman correlation, $r = -0.2886$, $p = 0.0421$, 95% confidence interval ($-0.5309$ to $-0.002725$). Data were presented as mean ± SD. The statistical tests are two-sided unless otherwise specified. For (**a–c**), the data were analyzed by Friedman test. For (**e**), unpaired Student's $t$-test. (*$p < 0.05$, **$p < 0.01$, ***$p < 0.001$, ****$p < 0.0001$). Source data are provided as a Source data file.

## circEZH2_005 protects intestinal organoids from H/R injury in vitro

Owing to the high expression of circEZH2_005 in Lgr5+ ISCs and the important role of ISCs in intestinal regeneration after I/R injury, we selected ISCs to evaluate the function of circEZH2_005 in vitro. Small interfering RNAs (siRNAs) and adenovirus were used to down- and upregulate the expression of circEZH2_005, respectively (Fig. 4a and Supplementary Fig. 6a). No significant changes were observed in Ezh2

expression (Fig. 4b and Supplementary Fig. 6b). Then, we selected the si-circEZH2_005-2 fragment to construct lentiviral vectors (shRNA) and infected stem cells for subsequent functional verification. circEZH2_005 deprivation aggravated the morphological and pathological injury of the organoid (Fig. 4c), repressed its vitality (Fig. 4d), and decreased the number of Ki67+ cycling cells, as well as BrdU+ cells, under H/R conditions (Fig. 4e). Changes in circEZH2_005 expression also reduced the levels of the tight junction proteins occludin and zonula occludens-1

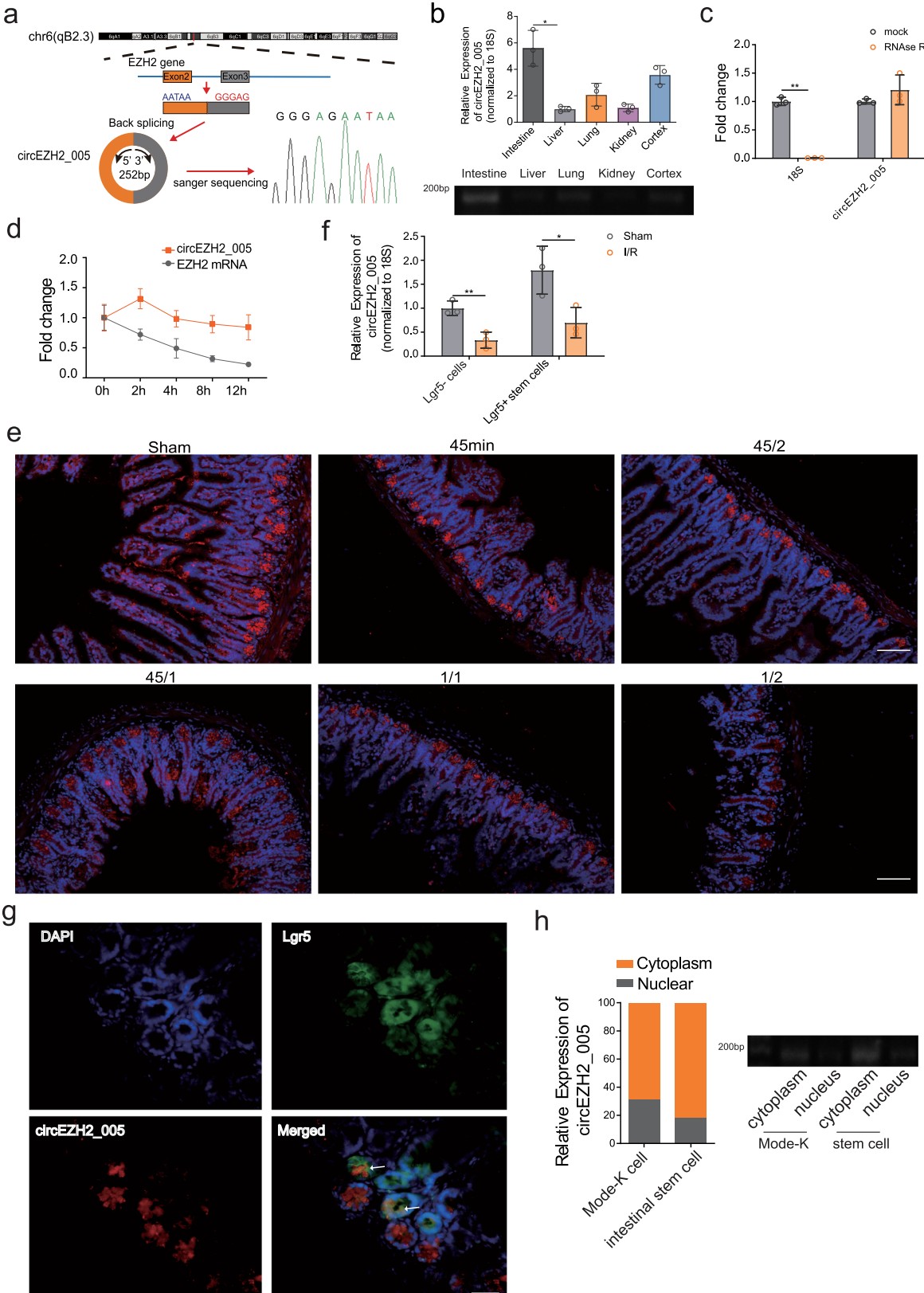

during H/R injury (Fig. 4f). circEZH2_005 depletion increased the expression of apoptotic proteins and decreased that of proliferation-related proteins induced by H/R (Supplementary Fig. 6f); circEZH2_005 overexpression had the opposite effect (Supplementary Fig. 6c–g). These results indicate that circEZH2_005 protects the proliferation of the intestinal crypt cell during H/R injury.

## circEZH2_005 ameliorates intestinal histopathological injury induced by I/R in vivo

Next, we evaluated the effect of circEZH2_005 on intestinal I/R injury in mice. We successfully overexpressed circEZH2_005 in the small intestine of mice by intraperitoneal injection of circEZH2_005-green fluorescent protein (GFP)-expressing adenovirus before I/R surgery

**Fig. 3 | Characterization and expression of circEZH2_005. a** Schematic illustration of circEZH2_005 formation through the circularization of exons 2 and 3 of EZH2 and the back-splice junction sequences were confirmed by Sanger sequencing. **b** RT-qPCR analysis of circEZH2_005 expression in the mouse organ tissue of intestinal, liver, lung, kidney, and brain (*n* = 3 mice per group). intestinal vs liver, *p* = 0.0257. **c** RT-qPCR analysis of the abundance of circEZH2_005 and 18 S mRNA in enterocytes treated with or without RNase R (*n* = 3 biological replicates). 18 S Mock vs 18 S RNase R, *p* = 0.0017. **d** RT-qPCR analysis of circEZH2_005 and EZH2 abundance in enterocytes treated with Actinomycin D at a consecutive time point. **e** Fluorescence in situ hybridization (FISH) experiments tested the localization of circEZH2_005 in the intestinal tissues of mice. The nuclei were stained with DAPI. The scale bar is 100 μm (*n* = 5 mice per group). **f** RT-qPCR analysis of circEZH2_005 expression in Lgr5+ crypt stem cells and Lgr5- cells after intestinal I/R treatment (*n* = 3 biological replicates). Lgr5- sham vs I/R, *p* = 0.0073. Lgr5+ sham vs I/R, *p* = 0.0330. **g** FISH assay showing that circEZH2_005 is colocalized with Lgr5 protein in Lgr5-GFP mice intestinal tissue. Scale bar is 50 μm. **h** RT-qPCR analysis of the relative quantity of circEZH2_005 in the cytoplasmic and nuclear fractions of enterocyte and intestinal crypt stem cells (*n* = 3 biological replicates). Data were presented as mean ± SD. For a,b, and f, the data were analyzed by the unpaired two-tailed Student's *t*-test. (*$p < 0.05$, **$p < 0.01$, ***$p < 0.001$, ****$p < 0.0001$). Source data are provided as a Source data file.

(Supplementary Fig. 7a and b). circEZH2_005 overexpression improved the I/R-induced decrease in the height of the villus and depth of the crypt (Fig. 5a). Inflammatory cytokine levels were also significantly lower in the I/R+circEZH2_005 group compared with those in the I/R+vector group (Fig. 5b). circEZH2_005 overexpression reversed the I/R-induced reduction of transit-amplifying cells (Ki67+ cells) and S-phase cells (BrdU+ -intestinal cells) (Fig. 5c). The restoration of circEZH2_005 significantly ameliorated the I/R-induced reduction in Olfm4 (another ISC marker) and Lgr5 expression in the jejunal crypts (Fig. 5d, e). Collectively, these results demonstrate that circEZH2_005 can prevent intestinal I/R injury by initiating ISC proliferation.

### hnRNPA1 binds to circEZH2_005 and promotes its cytoplasmic translocation

We investigated the underlying mechanism by which circEZH2_005 regulates ISCs. Via the analysis on the circRNADb website, we found that circEZH2_005 could not translate proteins (Supplementary Fig. 8a). Studies have confirmed that circRNAs interact with numerous proteins and participate in disease regulation[31,32]. Through RNA pull-down and proteomic analysis, we identified 309 proteins that specifically interact with circEZH2_005 (Fig. 6a). Gene Ontology enrichment analysis revealed that these proteins were associated with mRNA translation or stability (Supplementary Fig. 8b). Therefore, we speculated that the proliferative effect of circEZH2_005 on ISCs could be related to the regulation of RNA metabolism. The top six proteins with the highest scores or abundance were selected as candidates for validation.

hnRNPA1 was among the top six proteins with the highest scores in the catRAPID (http://s.tartaglialab.com/page/catrapid_omics_group), RBPDB (http://rbpdb.ccbr.utoronto.ca/), and RBPmap (http://rbpmap.technion.ac.il) database predictions (Fig. 6b and Supplementary Fig. 8c, d) and/or a high expression with the circEZH2_005-sense probe (Fig. 6c). In addition, hnRNPA1 was validated as an RBP candidate for circEZH2_005 by RNA immunoprecipitation (RIP) (Fig. 6d). Furthermore, the regulatory role of circEZH2_005 with regard to hnRNPA1 was observed when hnRNPA1 was localized in the nucleus in response to I/R stimulation (Fig. 6e). hnRNPA1 shuttles between the nucleus and cytoplasm in response to cellular stress[33]. We found that circEZH2_005 knockdown reduced the cytoplasmic level of hnRNPA1, whereas circEZH2_005 overexpression promoted the export of the hnRNPA1 protein from the nucleus to the cytoplasm (Fig. 6f and Supplementary Fig. 8e). This result was verified by immunofluorescence of circEZH2_005 overexpression (Supplementary Fig. 8f). The full-length hnRNPA1 has two functional RNA binding domains. RIP assays revealed that the RNA recognition motif 2 (RRM2) domain of hnRNPA1 was necessary for the binding of circEZH2_005 (Fig. 6g). RNA pull-down results showed that the hnRNPA1 with the truncated RRM2 region could not be pulled down by circEZH2_005 probes (Fig. 6h). The 171–252 nt segment of circEZH2_005 was discovered to be sufficient in binding hnRNPA1 via fragment mapping using biotin-labeled circEZH2_005 sequences (Fig. 6i).

Next, we interfered with hnRNPA1 in circEZH2_005 overexpressed cells and followed with functional analysis. Downregulating hnRNPA1 expression aggravated organoid damage induced by H/R and reversed the protective effect of circEZH2_005 overexpression in vitro (Fig. 6j, k). Finally, we overexpressed circEZH2_005 adenovirus lacking the hnRNPA1-binding region (circEZH2_005 MUT) in ISCs. The capacity of organoid vitality (Supplementary Fig. 8g) and cell proliferation was almost abrogated in circEZH2_005 MUT overexpressed cells (Supplementary Fig. 8h), suggesting that the interaction of circEZH2_005 with hnRNPA1 is necessary for the intestinal crypt cell proliferation during H/R injury.

### circEZH2_005 prevents intestinal I/R injury by stabilizing G*prc5a*

hnRNPA1 plays a critical role in regulating mRNA synthesis, stability, and translation[34]. Therefore, we evaluated intestinal crypt cells overexpressing circEZH2_005 by RNA sequencing. Overall, 717 mRNAs were significantly upregulated (fold change > 2.0; Fig. 7a). Among these, Gprc5a was most upregulated and predicted to bind to hnRNPA1, according to starBase data (https://starbase.sysu.edu.cn/starbase2/) (Fig. 7b, c and Supplementary Fig. 9a). The positive regulation of Gprc5a by circEZH2_005 was confirmed by western blotting (Fig. 7d). We then verified that silencing Gprc5a-aggravated organoids damaged by H/R functionally abrogated the organoids' protective effect of circEZH2_005 against H/R injury (Fig. 7e, f). Therefore, we selected Gprc5a for further analysis.

An important function of cytoplasmic hnRNPA1 is to stabilize target mRNAs by binding to their 3′-untranslated region (UTR)[35]. The 3′-UTR of the Gprc5a mRNA contains several predicted hnRNPA1 binding motifs (Supplementary Fig. 9b, c), and our RIP assays showed that Gprc5a mRNA was significantly enriched in anti-hnRNPA1 compared with IgG (Fig. 7g). As expected, silencing hnRNPA1 decreased the stability of Gprc5a mRNA, whereas the opposite results were observed when hnRNPA1 was overexpressed (Fig. 7h and Supplementary Fig. 9d). Furthermore, the deficiency or overexpression of hnRNPA1 affected the protein levels of Gprc5a (Fig. 7i). Furthermore, we found that circEZH2_005 knockdown decreased the binding of Gprc5a mRNA to hnRNPA1 (Fig. 7j). Consistently, the RNA pull-down assays showed that the overexpression or knockdown of circEZH2_005 increased or decreased the enrichment of hnRNPA1 bound to the 3′-UTR of Gprc5a mRNA, respectively (Fig. 7k). Next, circEZH2_005 regulated the stability of Gprc5a mRNA, and rescue experiments demonstrated that this effect was reversed by hnRNPA1 (Fig. 7l and Supplementary Fig. 9e). Correspondingly, hnRNPA1 silencing prevented the upregulation of Gprc5a protein levels upon circEZH2_005 overexpression (Supplementary Fig. 9f).

By constructing a mutant lentivirus of hnRNPA1 with the truncated RRM2 region, we further determined whether circEZH2_005 regulates the expression of Gprc5a and exerts protective effects by binding and transporting hnRNPA1 into the cytoplasm. We noticed that mutation of either circEZH2_005 or hnRNPA1 significantly reversed the hnRNPA1 cytoplasmic localization (Supplementary Fig. 9g), Gprc5a protein upregulation (Supplementary Fig. 9h), and organoid activity (Supplementary Fig. 9i) upon co-transfection of wild-type circEZH2_005 and wild-type hnRNPA1 lentivirus. Finally, we explored the effect of circEZH2_005 on Gprc5a expression in vivo.

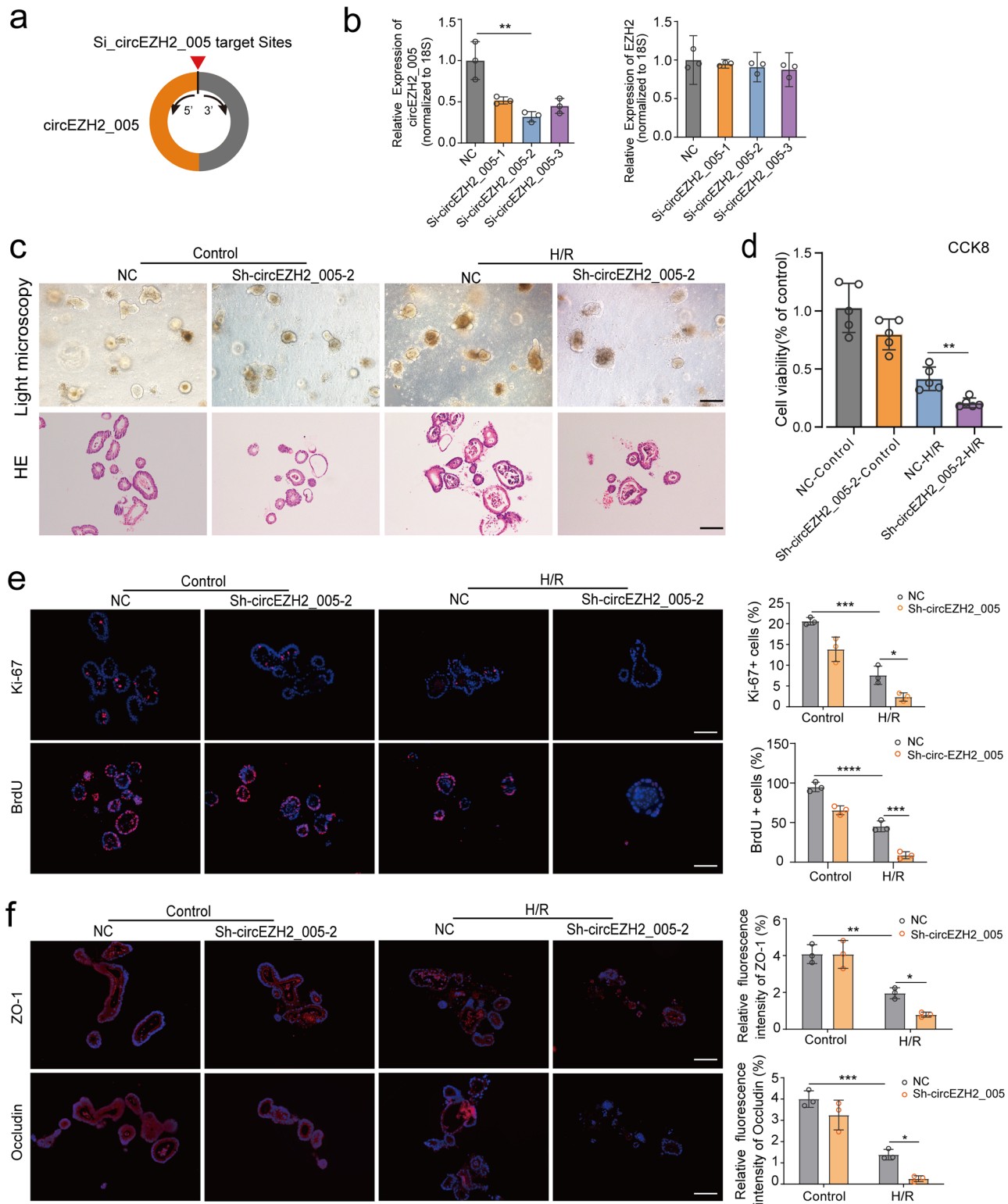

Gprc5a levels increased in intestinal crypt cells after circEZH2_005 overexpression in the I/R group, as shown by qRT-PCR and immuno-histochemical analyses (Fig. 7m, n). We conclude that circEZH2_005 improves the stability of Gprc5a by binding to hnRNPA1, which promotes ISC proliferation during intestinal I/R injury.

## Discussion

The clinical identification of acute intestinal injury is important given its associated high morbidity and mortality rates[36]. Early diagnosis of injured intestinal tissue is critical for determining the prognosis of patients with intestinal I/R injury. Despite the vast array of research in this field, an ideal biomarker for early diagnosis is yet to be identified. Here, we report that exosomal circEZH2_005 may serve as a valuable biomarker for the detection of intestinal I/R injury. Our animal and human studies revealed that circEZH2_005 expression in the plasma exosome is correlated with intestinal injury grade. Mechanistically, we demonstrate that circEZH2_005 directly interacts with hnRNPA1, thereby upregulating the expression of hypoxia-induced Gprc5a which

**Fig. 4 | CircEZH2_005 protects the intestinal organoid from H/R injury in vitro.**
**a** Schematic illustration showing the si-circEZH2_005 segment targeting the backspliced junction of circEZH2_005. **b** Intestinal crypt stem cells were transfected with three siRNAs specifically targeting circEZH2_005 (Si-1, Si-2, Si-3), and the level of circEZH2_005 and EZH2 were detected by qRT-PCR (n = 3 biological replicates). NC vs Si-2, $p = 0.0078$. **c** The organoid morphology was observed by light microscopy and HE staining, the scale bar is 100 μm (n = 3 biological replicates). **d** The organoid viability was analyzed by CCK-8 (n = 5 biological replicates). NC-H/R vs Sh-circEZH2_005-H/R, $p = 0.0079$. **e** Immunofluorescence staining for the Ki-67 and BrdU in the organoids for proliferation analysis, scale bar is 100 μm (n = 3 biological replicates). Ki-67, Control NC vs H/R NC, $p = 0.0001$. H/R NC vs H/R Sh-circEZH2_005-2, $p = 0.0285$; BrdU, Control NC vs H/R NC, $p < 0.0001$. H/R NC vs H/R Sh-circEZH2_005-2, $p = 0.0002$. **f** Immunofluorescence staining for the Occludin and ZO-1 protein expression in the organoids (n = 3 biological replicates). Zo-1, Control NC vs H/R NC, $p = 0.0018$. H/R NC vs H/R Sh-circEZH2_005-2, $p = 0.0459$; Occludin, Control NC vs H/R NC, $p = 0.0002$. H/R NC vs H/R Sh-circEZH2_005-2, $p = 0.0287$. Data were presented as mean ± SD. The statistical tests are two-sided unless otherwise specified. For (**b**), the data were analyzed by unpaired two-tailed Student's t-test. For (**d**), Mann–Whitney test. For (**e**) and (**f**), one-way ANOVA with Dunnett's test. (*$p < 0.05$, **$p < 0.01$, ***$p < 0.001$, ****$p < 0.0001$). Source data are provided as a Source data file.

protects intestinal crypt cell proliferation at I/R injury (Fig. 8). Consequently, our data raise the possibility that circEZH2_005 may be a useful therapeutic target for rescuing the ischemic intestinal tissue in I/R injury.

A hypoxic environment can change the type and quantity of secreted exosomes and their protein and nucleic acid content[37,38]. In our previous research, we successfully extracted exosomes and found that the quantity of nucleic acids, including microRNAs, in intestinal tissue exosomes changed significantly under I/R conditions in vivo[39,40] Therefore, we hypothesized that detecting changes in intestinal exosome content after intestinal I/R might help identify an ideal marker of intestinal injury. Patients who experience intestinal injury often develop other organ injuries, which can present significant confounders that may affect experimental results. Therefore, we used intestinal tissue exosomes from intestinal I/R mice for circRNA sequencing rather than plasma from patients, which is a limitation of this study. To avoid cells polluting intestinal tissue exosomes, we combined ultra-high speed and density-gradient centrifugation techniques, which allowed us to obtain exosome samples of high purity with a relatively low concentration. Additionally, to remove the influence of linear RNA on exosomes, we removed these biomolecules by enzymatic digestion before sequencing. These are the main reasons for the small number of differentially expressed circRNAs obtained from the subsequent sequencing results.

Through sequencing and screening in mouse intestinal I/R models, we obtained the target exosome circRNA circEZH2_005; to distinguish it from other Ezh2-derived circRNAs reported in the literature, we abbreviated it as circEZH2_005 according to the naming method in the CircBank database (hsa_circEZH2_005). We demonstrated that circEZH2_005 expression is significantly downregulated in the ischemic intestinal tissue and the plasma exosomes of mice with I/R. Moreover, plasma exosomal circEZH2_005 expression was negatively correlated with Chiu's score and IFABP level. Therefore, blood circEZH2_005 might be a highly specific predictive biomarker for intestinal injury in patients with intestinal I/R. We then explored the source of circEZH2_005 in intestinal tissue. Intriguingly, FISH revealed that circEZH2_005 was expressed in the whole intestinal mucosal epithelium but was predominantly located in the jejunal crypts. circEZH2_005 expression in the villous epithelium decreased considerably with the aggravation of the intestinal mucosa injury. Therefore, we suspect circEZH2_005 is likely derived from intestinal crypt cells and then delivered to intestinal mucosal epithelial cells and circulating plasma through exosomes.

The occurrence of gastrointestinal dysfunction is common during the perioperative period of CPB surgery[41]. One study has found that IFABP correlates positively with gastrointestinal complications in patients who developed GI dysfunction after cardiac surgery[42]. Based on this knowledge, we used cardiac surgery as the intestinal I/R model for further analysis. We selected patients undergoing the same type of operation (including coronary artery bypass grafting or aortic valve replacement) while excluding patients who already had gastrointestinal diseases before surgery. Analysis of their plasma revealed that circEZH2_005 was significantly downregulated immediately after

surgery. Because the diagnosis of intestinal injuries is time-consuming and the GI was initiated intraoperatively, we selected the time point immediately after the operation to perform receiver operating characteristic curve and correlation analyses. The AGI grade on the first day after surgery was also determined, but whether the expression of circEZH2_005 at other time points is of diagnostic significance needs future research. We selected AGI grade II as the outcome criterion for intestinal I/R injury[43]. Although the AUC value of circEZH2_005 to distinguish intestinal I/R injury was relatively low (0.732), it increased the diagnostic efficacy of the common enterocyte injury marker IFABP from an AUC of 0.776 to 0.846, reflecting a good clinical application value. GI plays a crucial role in the progression of infective complications and multiple organ dysfunction syndromes after an operation, which leads to a poor prognosis in cardiac surgery patients[44]. Despite the clear evidence that the serum IFABP level was an objective predictor of prognosis in postoperative cardiac surgery patients, we believe that it will be better in combination with circEZH2_005 to predict outcomes of cardiac surgery patients and even other critical illnesses, such as septic shock, acute cardiac arrest, and acute pancreatitis, to reduce the duration of ICU stay and improve the prognosis[41]. This is a pertinent topic for future studies. Given the relatively small sample size and single-center design of this study, our results are a preliminary evaluation of whether circEZH2_005 can distinguish patients with intestinal I/R injury from those without. These results need to be validated in a larger multicenter cohort to evaluate the diagnostic performance of circEZH2_005 before its translation into routine clinical practice. Nevertheless, our research still demonstrates that circEZH2_005 may be a potential biomarker that was related to the severity of intestinal I/R injury.

Considering that circEZH2_005 was highly expressed in jejunal crypts and Lgr5+ ISCs, we wondered whether circEZH2_005 also played a regulatory role in the pathophysiological process of intestinal I/R by activating ISCs. The functional experiments showed that circEZH2_005 could ameliorate the impaired crypt cell proliferation ability induced by I/R and restore the decrease in villus height and crypt depth, which indicated that circEZH2_005 might protect I/R injury at least in part by improving the proliferative ability of crypt cells. Nevertheless, the role of circEZH2_005 in other intestinal epithelial cells remains to be further investigated.

circRNAs perform various biological functions, including interacting with RBP[45]. Therefore, we investigated the binding protein of circEZH2_005. Combining bioinformatic predictions and mass spectrometry, we selected hnRNPA1 as the RBP of circEZH2_005 in ISCs. Our results demonstrated that circEZH2_005 can directly bind to hnRNPA1 in ISCs. hnRNPA1 improves intestinal injury in enteric mice by enhancing epithelial restoration[46]. hnRNPA1, an RBP involved in the RNA maturation process in the nucleus, is involved in the stabilization and translation of mRNAs in the cytoplasm[34]. The cytoplasmic accumulation of hnRNPA1 is a consequence of cellular stress response[35]. Hypertonic stress-induced cytoplasmic accumulation of hnRNPA1 regulates the internal ribosome entry site-mediated expression of Bcl-xL[47]. Herein, we report that upon I/R stimuli, hnRNPA1 accumulates mainly in the nucleus, accompanied by a decrease in circEZH2_005

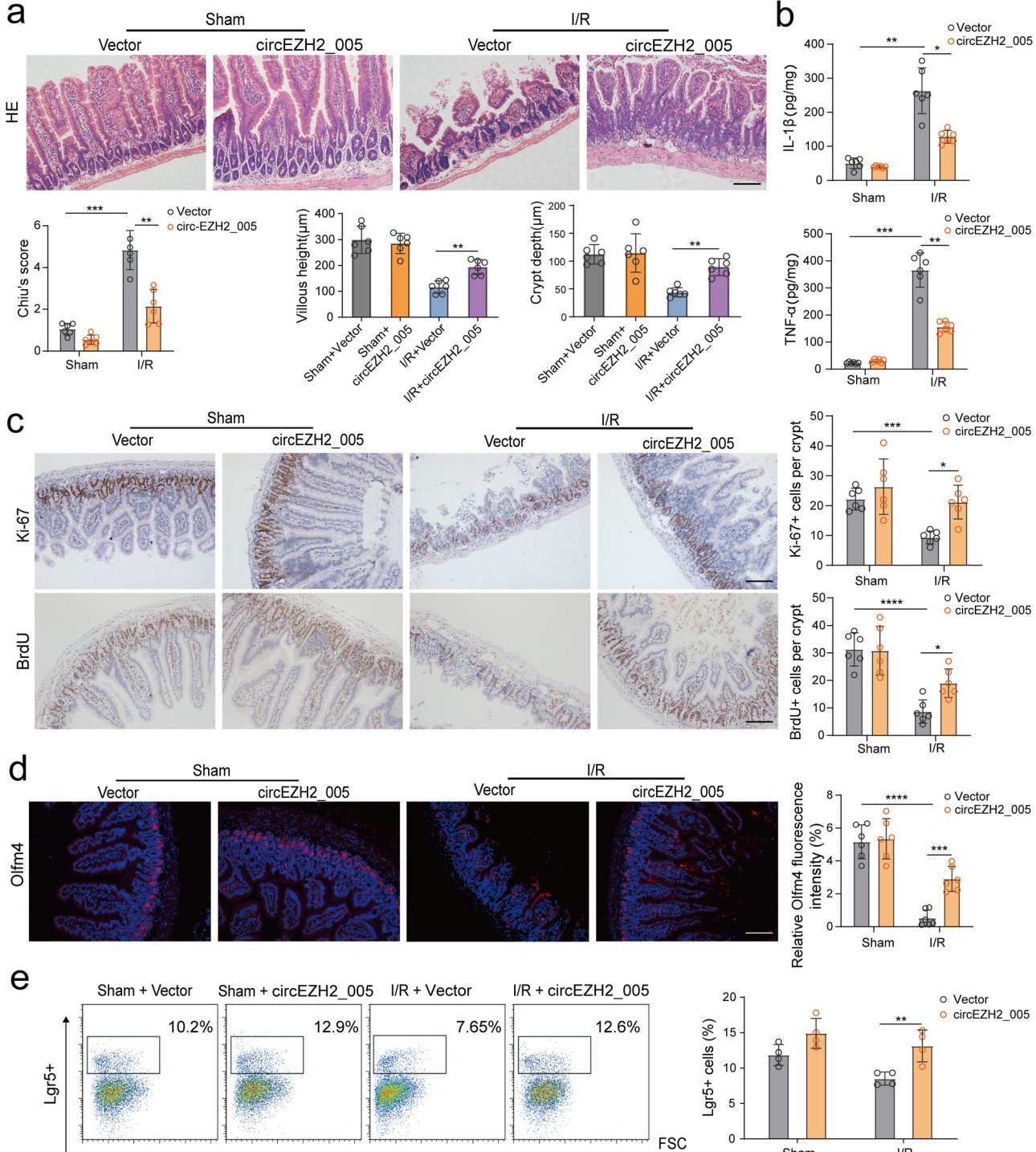

**Fig. 5 | circEZH2_005 ameliorates intestinal histopathological injury induced by I/R in vivo. a** Representative images of intestinal morphology HE staining showing the villous height and crypt depth, scale bar is 100 μm (*n* = 6 mice per group). Chiu's score, Sham Vector vs I/R Vector, *p* = 0.0004. I/R Vector vs I/R cir-cEZH2_005, *p* = 0.0018; Villous height, I/R Vector vs I/R circEZH2_005, *p* = 0.0054; Crypt depth, I/R Vector vs I/R circEZH2_005, *p* = 0.0037. **b** ELISA analysis of cir-cEZH2_005 overexpression on the influence of the expression of intestinal tissue inflammatory factors IL-1β and TNF-α (*n* = 6 mice per group). IL-1β, Sham Vector vs I/R Vector, *p* = 0.0014. I/R Vector vs I/R circEZH2_005, *p* = 0.0153; TNF-α, Sham Vector vs I/R Vector, *p* = 0.0002. I/R Vector vs I/R circEZH2_005, *p* = 0.0012. **c** Immunohistochemistry staining of the Ki-67 and BrdU in the intestinal tissue for proliferation analysis, scale bar is 100 μm (*n* = 6 mice per group). Ki-67, Sham

Vector vs I/R Vector, *p* = 0.0006. I/R Vector vs I/R circEZH2_005, *p* = 0.0148; BrdU, Sham Vector vs I/R Vector, *p* < 0.0001. I/R Vector vs I/R circEZH2_005, *p* = 0.0270. **d** Immunofluorescence staining for the Olfm4 in the intestinal tissue, scale bar is 100 μm (*n* = 6 mice per group). Sham Vector vs I/R Vector, *p* < 0.0001. I/R Vector vs I/R circEZH2_005, *p* = 0.0007. **e** Flow cytometry analysis of the effect of cir-cEZH2_005 overexpression on the influence of the number of Lgr5 positive cells (*n* = 4 mice per group). I/R Vector vs I/R ircEZH2_005, *p* = 0.0084. Data were pre-sented as mean ± SD. The statistical tests are two-sided unless otherwise specified. The data were analyzed by the one-way ANOVA with Dunnett's test. (**p* < 0.05, ***p* < 0.01, ****p* < 0.001, *****p* < 0.0001). Source data are provided as a Source data file.

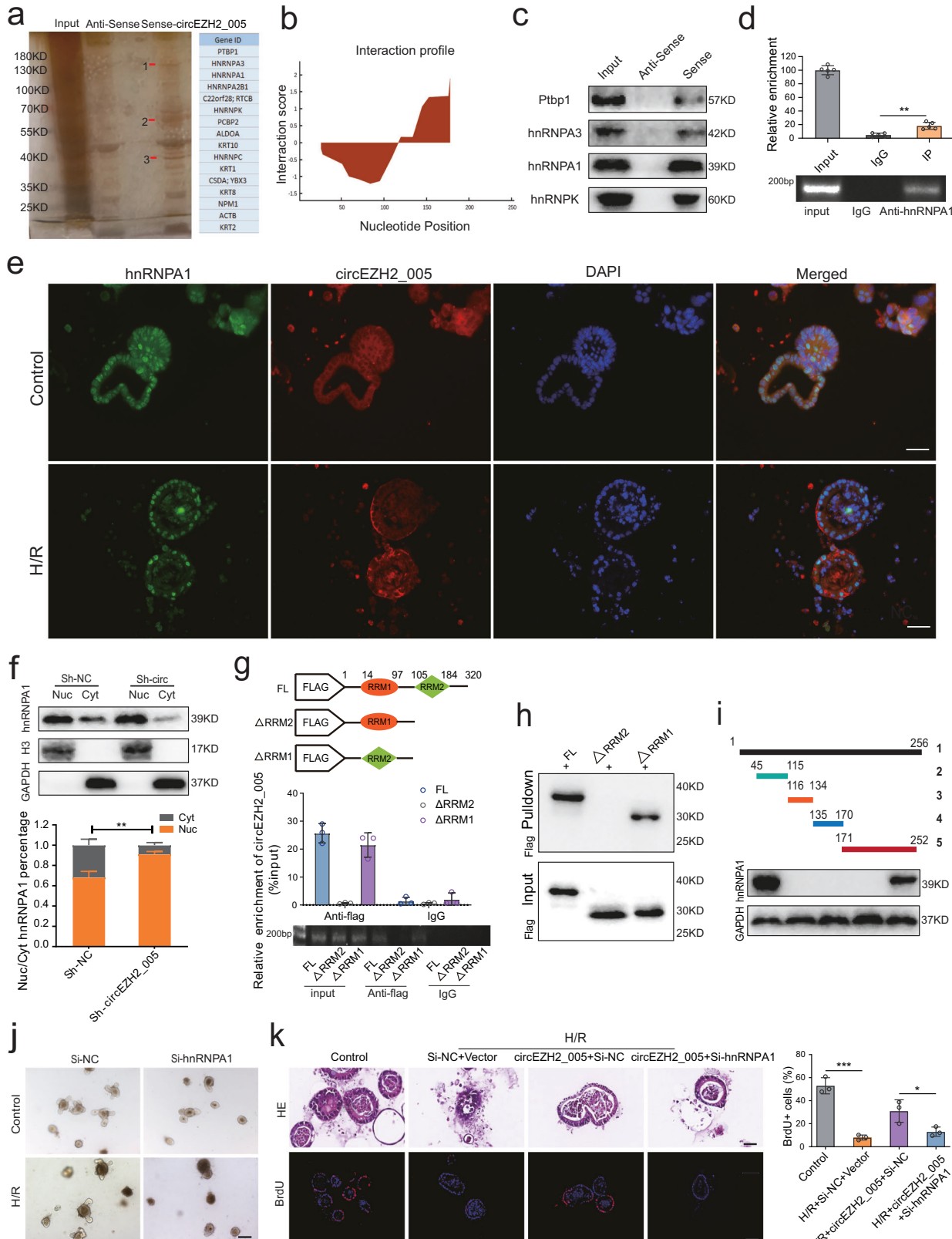

expression. Therefore, we speculated whether circEZH2_005 is involved in the nucleoplasmic transport of hnRNPA1 under I/R conditions. This hypothesis is supported by our data showing that circEZH2_005 promoted the export of the hnRNPA1 protein from the nucleus to the cytoplasm in the presence of H/R injury.

Transcriptome sequencing and bioinformatics analyses showed that most genes differentially expressed upon circEZH2_005 over-expression (including Gprc5a) can bind to hnRNPA1. Thus, we speculated that circEZH2_005 directly binds to hnRNPA1 to regulate the mRNA expression profile of ISCs and, consequently, their behavior.

**Fig. 6 | HnRNPA1 binds to circEZH2_005 in crypt stem cells and promotes its cytoplasmic translocation. a** RNA pull-down assay was performed using a biotinylated circEZH2_005 probe, followed by mass spectrometry. Diferential band to bind circEZH2_005 was identifed. **b** The binding sites between circEZH2_005 and hnRNPA1 were predicted by catRAPID. **c** Western blot analysis of RBPs in pulldown assays targeting the junction of circEZH2_005. **d** Association between circEZH2_005 and hnRNPA1 was used for RIP assay (n = 6 biological replicates). IgG vs IP, p = 0.0033. **e** IF-FISH assay showing that circEZH2_005 is colocalized with hnRNPA1 protein in the cytoplasm, scale bar is 50 μm. **f** Western blotting to check the expression of cytoplasmic (Cyt) and nuclear (Nuc) hnRNPA1 in circEZH2_005-silenced intestinal crypt stem cells (n = 3 biological replicates). Cyt, Sh-NC vs Sh-circEZH2_005, p = 0.0013. Nuc, Sh-NC vs Sh-circEZH2_005, p = 0.0013. **g** The full-length or truncated forms of flag-labeled hnRNPA1 plasmids were constructed, and then RIP assay was performed to detect whether the truncated hnRNPA1 can pull

down circEZH2_005 (n = 3 biological replicates). **h** Different domains of hnRNPA1 protein were incubated with circEZH2_005-specific probes, followed by an RNA pull-down assay. **i** Truncated fragments of biotinylated circEZH2_005 were incubated with crypt cell lysates, followed by an RNA pull-down assay. **j** Ileum organoid morphology was observed by light microscopy, the scale bar is 100 μm. **k** Ileum organoid morphology was observed by HE staining and the expression of BrdU in the organoids was observed by immunofluorescence, the scale bar is 50 μm and 100 μm (n = 3 biological replicates). Control vs H/R + Si-NC + Vector, p = 0.0001. H/R + circ-EZH2_005 + Si-NC vs H/R + circ-EZH2_005 + Si-hnRNPA1, p = 0.0373. Data were presented as mean ± SD. The statistical tests are two-sided unless otherwise specified. The data were analyzed by the one-way ANOVA with Tukey's test. (*p < 0.05, **p < 0.01, ***p < 0.001, ****p < 0.0001). Source data are provided as a Source data file.

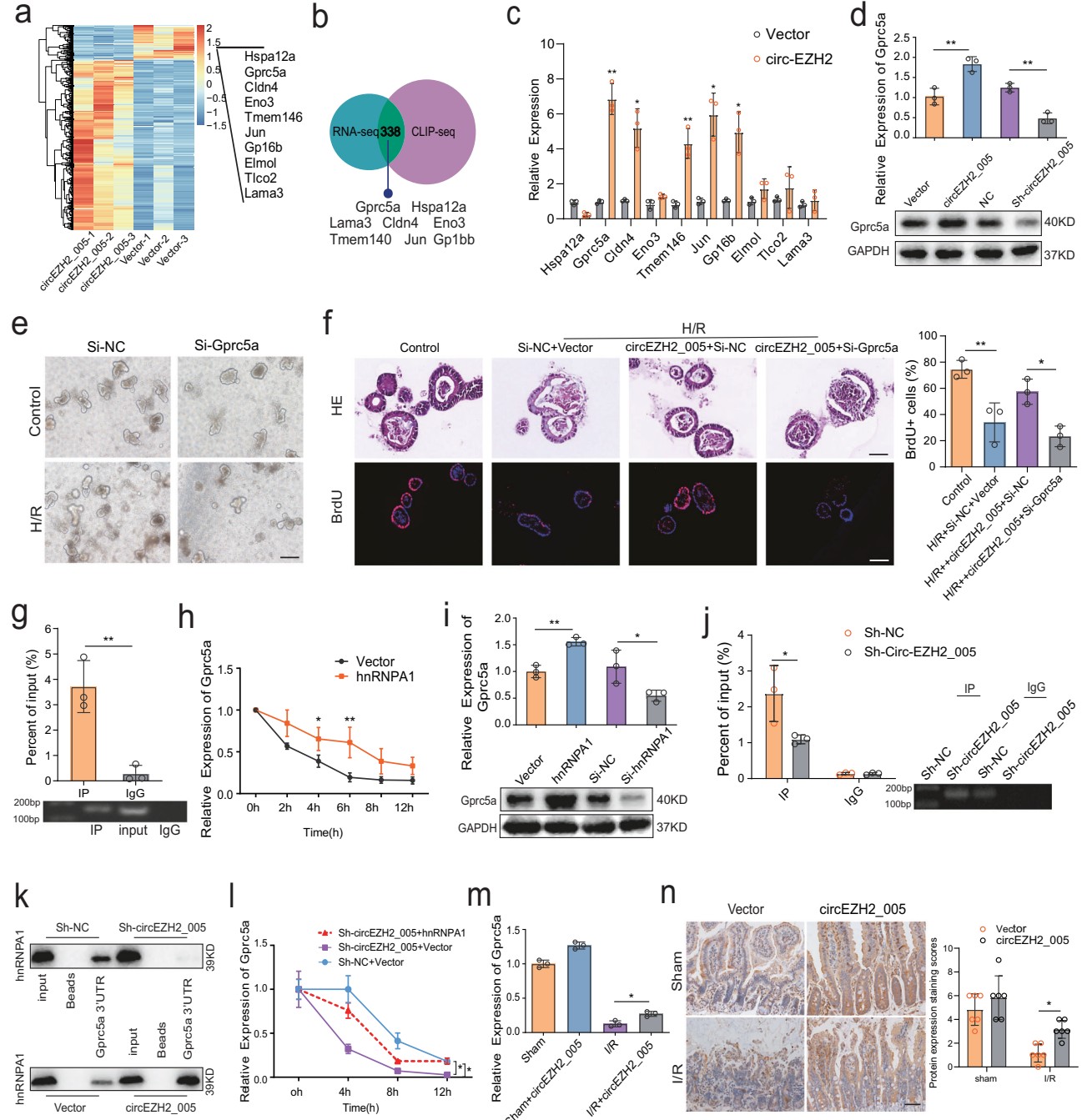

**Fig. 7 | CircEZH2_005 prevents intestinal I/R injury by stabilizing G*prc5a*.**
**a** Heatmap of differentially expressed mRNAs in circEZH2_005-upregulated and control intestinal crypt cells. **b** Venn diagram showing the intersection of target genes between transcriptome sequencing results and predicted target genes of hnRNPA1. **c** Expression levels of top 10 upregulated mRNAs in circEZH2_005-upregulated cells (*n* = 3 biological replicates). Gprc5a, *p* = 0.0069; Cldn4, *p* = 0.0225; Tmem146, *p* = 0.0023; Jun, *p* = 0.0184; Gp1bb, *p* = 0.0283. **d** Western blot showing the protein level of Gprc5a after intervening in the expression level of circEZH2_005 (*n* = 3 biological replicates). Vector vs circ-EZH2_005, *p* = 0.0072. NC vs Sh-circ-EZH2_005, *p* = 0.0015. **e** Ileum organoid morphology was observed by light microscopy, the scale bar is 100 μm (*n* = 3 biological replicates). **f** Ileum organoid morphology was observed by HE staining and the expression of BrdU in the organoids was observed by immunofluorescence, the scale bar is 50 μm and 100 μm (*n* = 3 biological replicates). Control vs H/R + Si-NC + Vector, *p* = 0.0057. H/R + circ-EZH2_005 + Si-NC vs H/R + circ-EZH2_005 + Si-Gprc5a, *p* = 0.0154. **g** RIP assays show the association of hnRNPA1 with Gprc5a (*n* = 3 biological replicates). IgG vs IP, *p* = 0.0052. **h** The rate of degradation of the G*prc5a* was assessed after hnRNPA1 overexpressing (*n* = 5 biological replicates). 4 h, Vector vs hnRNPA1, *p* = 0.0238. 6 h, Vector vs hnRNPA1, *p* = 0.0012. **i** Western blot assay examined the protein level of Gprc5a after intervening in the expression of hnRNPA (*n* = 3

biological replicates). Vector vs hnRNPA1, *p* = 0.0024. NC vs Si-hnRNPA1, *p* = 0.0454. **j** RIP assay of the enrichment of Gprc5a with hnRNPA1 between the circEZH2_005 downregulated and NC group in intestinal crypt cells (*n* = 3 biological replicates). IP Sh-NC vs IP Sh-circEZH2_005, *p* = 0.0107. **k** RNA pull-down assays were performed in intestinal crypt cells with biotin-labeled Gprc5a 3′UTR. **l** The degradation rate of Gprc5a mRNA in intestinal crypt stem cells was evaluated after circEZH2_005 interference and hnRNPA1 overexpression simultaneously (*n* = 3 biological replicates). Sh-circ-EZH2_005 + hnRNPA1 vs Sh-circ-EZH2_005 + Vector, *p* = 0.0273. Sh-circ-EZH2_005 + Vector vs Sh-NC + Vector, *p* = 0.0109. **m** RT-qPCR analysis of Gprc5a expression in the mice intestinal I/R mode after circEZH2_005 overexpressing (*n* = 3 mice per group), Sham vs Sham + circ-EZH2_005, *p* = 0.0034. I/R vs I/R + circ-EZH2_005, *p* = 0.0132. **n** Immunohistochemical analysis of Gprc5a expression in the mice intestinal I/R mode after circEZH2_005 overexpressing, scale bar is 50 μm (*n* = 6 mice per group). I/R vs I/R + circ-EZH2_005, *p* = 0.0311. Data were presented as mean ± SD. The statistical tests are two-sided unless otherwise specified. For (**c**, **d**, **g**–**i**), the data were analyzed by the two-tailed unpaired Student's *t*-test. For (**f**), one-way ANOVA with Tukey's test. For (**j**–**n**), one-way ANOVA with Dunnett's Test. (**p* < 0.05, ***p* < 0.01, ****p* < 0.001, *****p* < 0.0001). Source data are provided as a Source data file.

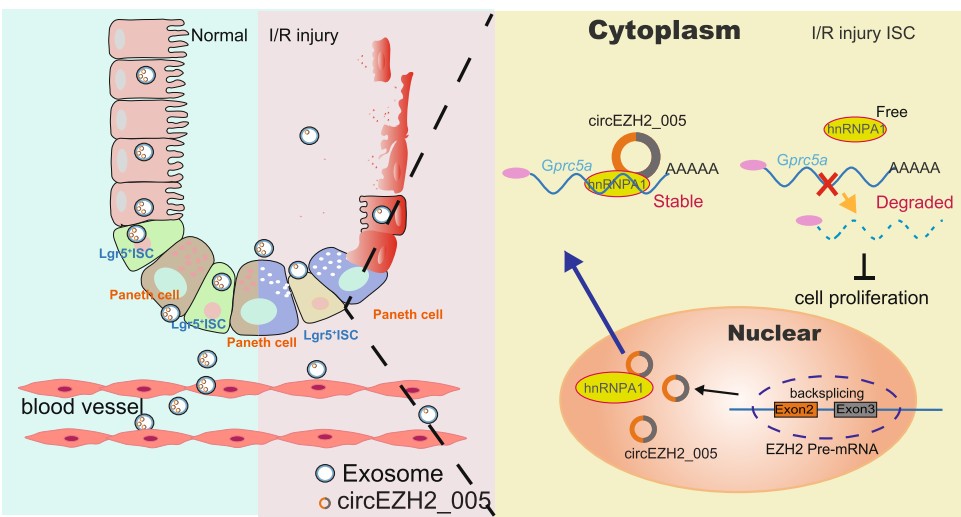

**Fig. 8 | Schematic illustration of the mechanism whereby circEZH2_005 regulates G*prc5a* stabilizing via hnRNPA1.** In general, circEZH2_005 is mainly expressed in intestinal crypt cells. After I/R injury, the expression of circEZH2_005 is decreased in the Lgr5+ stem cells and plasma exosomes. Meanwhile, the reduced expression of circEZH2_005 results in decreased hnRNPA1 binding and cytoplasmic export, thus inhibiting the binding of hnRNPA1 to G*prc5a*. Therefore, G*prc5a* becomes free and is degraded in the cytoplasm, which may alter the expression of Gprc5a-regulated proliferation genes. Consequently, reduced circEZH2_005 expression aggravates the I/ R-induced Lgr5+ stem cell injury, contributing to the development and progression of intestinal I/R injury.

The cytoplasmic hnRNPA1-mRNA interaction has functional relevance in mRNA stability. hnRNPA1 may regulate mRNA stability owing to its ability to bind to AUUUA-rich sequences[48]. Consistently, we revealed that circEZH2_005 facilitates hnRNPA1 cytoplasmic accumulation, thereby enhancing the stability of Gprc5a mRNA. Gprc5a plays a crucial role in the antihypoxic damage of cells[49]. Indeed, Gprc5a has been confirmed as a functional Hippo/YAP signaling activator protein in response to extracellular hypoxic signals[50,51]. Hippo/YAP signaling is a key pathway in Lgr5+ stem cell regulation during intestinal tissue regeneration[52,53]. Based on this information, we selected Gprc5a as the downstream regulatory protein of circEZH2_005. We demonstrated that the lack of Gprc5a exacerbated organoids involved in H/R injury in vitro and confirmed that circEZH2_005 positively regulated Gprc5a expression in vivo. However, we did not further evaluate the role of Gprc5a in the proliferative effect of circEZH2_005 in vivo, which is also a limitation of our study.

In summary, this study describes an intestinal I/R injury-associated circRNA, termed circEZH2_005, which was reliably detected in plasma exosomes and could effectively differentiate patients

with intestinal I/R injury from those without. Furthermore, circEZH2_005 could alleviate intestinal mucosal injury by facilitating crypt stem cell proliferation through direct interaction with hnRNPA1 and enhancing the stability of Gprc5a during intestinal I/R. Therefore, this study not only provides a promising noninvasive biomarker for the early detection of intestinal I/R injury but also opens an avenue for therapeutic strategies to combat intestinal I/R injury.

## Methods

### Human plasma collection

Fifty patients who were assigned for CPB surgery between May 2021 and February 2022 at the Department of Cardiac Surgery, Southern Medical University Nanfang Hospital (Guangzhou, China) were recruited given their increased risk of intestinal I/R injury. In patients undergoing cardiac surgery, CPB is potentially responsible for intestinal ischemia (due to reduced blood supply and oxygen delivery) and injury; therefore, these patients were used as a clinical model of intestinal I/R samples[27,28]. Exclusion criteria were: (1) age <18 years or >75 years, (2) chronic kidney disease, and (3) chronic digestive system

disease, previous gastrointestinal surgery, or confirmed or suspected intestinal ischemia/necrosis. Blood samples were collected pre-operatively (T0) and 6 h (T1), 12 h (T2), and 24 h (T3) after surgery in tubes with ethylenediamine tetra-acetic acid. Serum was obtained by centrifuging the blood samples at $3000 \times g$ at 4 °C for 10 min and stored at −80 °C until further analysis.

The AGI grading system was used to assess intestinal injuries of critically ill patients[43]. Intestinal ischemia was determined using the AGI score during the first 3 postoperative days, and those with an AGI grade <II were assigned to the control group. Demographic and clinical information on the patients is provided in Supplementary Table 1. All experiments performed and protocols used were approved by the Ethical Committee of the Nanfang Hospital, Southern Medical University (approval number: NFEC-202009-k2-01), and informed consent was obtained from all patients.

### Establishment of I/R injury model and study protocol
C57BL/6 male mice (6–8 weeks old) were obtained from the animal center of Southern Medical University (Guangzhou, China). All animals were housed in a specific pathogen-free environment with controlled temperature (20–22 °C) and humidity (50–70%), a 12/12 h light/dark cycle, and free access to food and water. The chow of the mice is SPF clean grade (MD17121) (medicience, Jiangsu, China). All experimental procedures involving animals were performed according to the National Institutes of Health guidelines and were reviewed and approved by the Ethics Committee of the Nanfang Hospital of Southern Medical University (approval number: NFYY-2019-0754). The mouse intestinal I/R injury model was established as previously described[54]. In brief, all mice were fasted for 12 h before surgery and drank water freely. Then the mice were anesthetized with 3% isoflurane followed by maintenance at 1.5% isoflurane in oxygen inhalation. Buprenorphine (0.1 mg/kg) and ketamine (10 mg/kg) were injected subcutaneously to prevent pain before the operation. The mice were placed on their backs on an electric blanket and maintained at a constant temperature. The abdomen was sterilized in preparation for surgery. Then, a 1 cm midline abdominal incision was made, and the small intestine was exteriorized using cotton swabs moistened in saline to expose the superior mesenteric artery. The superior mesenteric artery was clamped with a microvascular clip. During the operation, normal saline was dripped to moisten the intestine and ensure that the surgical dressing was moist. After 60 mins of ischemia, the micro clip was removed to initiate blood reperfusion, and the abdominal wound was closed. Check mice at least every 30 min to ensure stability. Immediately after the abdominal wound was sutured, 1 mL of normal saline was injected subcutaneously into the abdominal wall. Mice were maintained in a heated clean cage for the reperfusion phase, and their breathing and heartbeat rates were closely observed. After the surgical procedure, the mice were resuscitated in a thermostatic resuscitation chamber and received postoperative analgesia. In case of blood collection necessity, pentobarbital was administered intraperitoneally for terminal anesthesia. If blood collection was not required, euthanasia of mice was performed through $CO_2$ inhalation in this study. Following euthanasia, it is imperative to verify if any mice survived.

The number of mice in the experiments was $n = 6$ in each group, and the investigator who established the intestinal I/R model was blinded to the experimental design and group allocation. The mice were randomly separated into four groups: Vector+sham, Ad-circEZH2_005+sham, Vector+I/R, and Ad-circEZH2_005 + I/R. All groups were injected with 0.1 mL of Ad-Vector or Ad-circEZH2_005 ($1 \times 10^9$ TU/mL) (GeneChem, Shanghai, China) intraperitoneally 4 days before the surgery. Except for not performing I/R operation, all other operation steps are the same between I/R and sham groups. There was no difference in the body weight and age of the mice. They were all 6–8-week-old C57BL/6 J mice weighing 20–23 g.

### Chiu's score
Chiu's score was used to evaluate the degree of intestinal injury. Intestinal damage was scored on a scale from 0 (normal) to 5 (damaged severely) as follows: 0−normal mucosal villi; 1−development of sub-epithelial Gruenhagen space usually at the apex of the villus, often with capillary congestion; 2−an extension of the subepithelial space with the moderate lifting of the epithelial layer from the lamina propria; 3−marked epithelial lifting down the sides of the villi, and possibly containing a few denuded tips; 4−denuded villi with exposed lamina propria and dilated capillaries, and possibly with increased cellularity in the lamina propria; and 5−digestion and disintegration of the lamina propria, hemorrhage, and ulceration. At least five randomly chosen fields from each mouse were evaluated and averaged to quantitate mucosal damage. Two independent pathologists blinded to the experimental groups evaluated damage to the intestinal mucosa.

### Lgr5-GFP+ISC isolation and organoid culture
Small intestinal organoids were isolated and cultured as previously described[55]. In brief, the small intestine was opened longitudinally and then washed in cold PBS with vigorous shaking, and the intestine was then cut into 2 mm pieces. The intestine samples were continuously washed in cold PBS with a pipette until the supernatant was clear. The samples were then incubated in 30 mM EDTA for 15 min at 4 °C and vortexed for 15 s to separate the crypts. Then the samples were filtered through a 70 µm cell strainer, followed by $200 \times g$ centrifugation for 5 min. After being washed in cold PBS, the crypts were collected in a 50 ml centrifuge tube. For ISC isolation, the crypts were digested in 0.1% type I collagenase (Invitrogen) and incubated in 1× TrypLE Express (Life Technologies) supplemented with $1 kU mL^{-1}$ DNase I (Worthington, USA) for single-cell preparation. Lgr5-GFP + ISC were classified using fluorescence-activated cell sorting (FACS) and incubated with IntestiCult medium (STEMCELL Technologies, Vancouver, Canada) for growing into organoids. To induce the organoid H/R model, organoids were placed in an anaerobic environment containing 5% $CO_2$, 2% $O_2$, and 93% $N_2$ in a 37 °C incubator for 4 h. Flow cytometry was performed using FACS Calibur instruments (BD Biosciences, San Jose, CA, USA), and data were analyzed using FlowJo (Tree Star Inc.).

### CircRNA sequencing
The total RNA of exosomes was extracted using the TRIzol method, and the extracted total RNA was sampled for quality inspection. The Agilent 2100 bioanalyzer was used to analyze the concentration and integrity of RNA. After passing the RNA quality inspection, 10 ng to 20 ng of RNA was digested with RNase R to remove linear RNA and used for library preparation. The RNA was fragmented, the 5' end conversion template was added, and the first strand of cDNA was reverse-transcribed. After the first round of PCR plus the index connector, the ligation product was purified, and the ribosomal cDNA was removed with ZapR and R-probes. The library was amplified by selecting the appropriate number of cycles according to the initial amount. After amplification, the circRNA SEQ library was purified and isolated with magnetic beads. After library quality control, different libraries were pooled according to the requirements for effective concentration and target offline data volume. The NovaSeq 6000 System with PE150 mode was used for sequencing. We defined statistical criteria for selecting differentially expressed circRNAs using |fold changes| ≥ 2.0 with $p < 0.05$. The sequencing was completed by Geneseed Biotech (Guangzhou, China).

### Hematoxylin and eosin staining
The ileum tissue samples were fixed in 4% paraformaldehyde, embedded in paraffin, and cut into 5-µm sections. The samples were

then stained with hematoxylin and eosin, as previously described[56]. The images were captured at 200× magnification using a fluorescence microscope (Olympus, Tokyo, Japan). Two pathologists blinded to the study groups independently evaluated and scored intestinal mucosal injury according to the Chiu scoring system.

## Detection of organoid injury

The CCK-8 assay (Dojindo Molecular Technologies, Rockville, USA) was used to detect cell viability and assess organoid damage according to the manufacturer's instructions. Briefly, cells were seeded at a density of $1 \times 10^4$ cells/well on 96-well plates. CCK8 solution (10 μL) was added to each well and incubated for 2 h at 37 °C, after which the absorbance was measured at 450 nm using a microplate reader (Thermo Fisher Scientific, Waltham, USA). Cell viability was determined based on the respective absorbance values compared to the control well, according to the manufacturer's instructions.

## Immunofluorescence and immunohistochemistry

Immunofluorescence and immunohistochemistry were performed as previously described[54]. For immunofluorescence, after antigen repair, the tissue sections were permeabilized with 0.3% Triton X-100 for 10 min and then blocked with 5% BSA at 37 °C for 1 h and incubated with primary antibody overnight at 4 °C. The next day, cells were washed with PBS and then incubated with the corresponding secondary antibody for 30 min at 37 °C, followed by staining with DAPI. For immunohistochemical, after antigen repair, endogenous peroxidases were inactivated in 3% $H_2O_2$ solution for 10 min and then incubated with the primary antibody overnight at 4 °C. Then, the tissue sections were rinsed and incubated with enzyme-labeled secondary antibodies (PV-6001; ZSGB-BIO, Beijing, China) for 45 min at room temperature and staining was revealed by a DAB reaction (ZLI-9017; ZSGB-BIO, Beijing, China). The tissues were counterstained with haematoxylin.

Anti-Rabbit ZO-1 (Cat #ab221547; Abcam) at 1:400 of dilution. Anti-Rabbit Occludin (Cat #ab216327; Abcam) at 1:400 of dilution. Anti-Ki-67 (ab279653; Abcam, Cambridge, UK; 1:500 in dilution), anti-BrdU (A1482; ABclonal, Wuhan, China; 1:600 in dilution), anti-hnRNPA1 (YT2192; ImmunoWay, Plano, USA; 1:300 in dilution), and anti-GPRC5A (ab188905; Abcam, USA; 1:500 in dilution) antibodies were used to detect protein expression in intestinal tissues and organoids. Alexa Fluor 594-conjugated secondary antibody (711-585-152, Jackson Immunoresearch) at 1:400 of dilution. a Mo IgG/Alexa Fluor 594-conjugated secondary antibody (ZF-0513, ZSGB-BIO) at 1:200 of dilution. Images were captured at 200× magnification with an immunofluorescence microscope (Olympus). The relative intensity of the protein staining was quantified through automated image analysis in five randomly chosen fields in each sample.

## RNA extraction, gene expression analysis, and genomic DNA extraction

Total RNA was extracted using Invitrogen TRIzol reagent (Thermo Fisher Scientific) and reverse-transcribed into cDNA for further analysis. QRT-PCR was performed using an ABI Q6 Real-Time PCR System (Thermo Fisher Scientific) with the SYBR Green detection protocol (TOYOBO, Osaka, Japan). The expression of target genes was normalized against that of the housekeeping gene *Rn18s* using the $2-\Delta\Delta CT$ method. The primers used in the study are shown in Supplementary Table S2. Genomic DNA (gDNA) was extracted from cells using the Easy Pure Genomic DNA kit (TransGen Biotech Co., Beijing, China).

## RNA immunoprecipitation

The RIP assay was performed using the EZ-Magna RIP Kit (#17–701; Merck, Burlington, USA) with hnRNPA1-specific antibodies following the manufacturer's instructions. Briefly, the cells were lysed in RIP lysis buffer on ice for 45 min. After centrifugation, the supernatant was

incubated with 30 μl of Protein-A/G agarose Beads and antibodies. After overnight incubation, the immune complexes were centrifuged and then washed with a washing buffer. The immunoprecipitated RNAs were extracted using TRIzol and analyzed by qRT-PCR. Total RNAs (input) and isotype antibodies (IgG) were used as controls.

## Exosome collection and characterization

Intestinal tissue exosomes were isolated, as previously described[39]. The murine intestinal epithelial MODE-K cell line (ID: BNCC338300) was purchased from BeNa Culture Collection. These cell lines were cultured in DMEM medium (Gibco) supplemented with 10% foetal bovine serum (Gibco) according to standard protocols. Cell exosomes were isolated from 40 mL of culture medium on ice. The culture medium was then centrifuged at $400 \times g$ for 30 min, followed by centrifugation at $2000 \times g$ for 20 min. The resulting supernatant was filtered using a 0.22-μm strainer (R9SA12882; Merck) and then centrifuged at $12,000 \times g$ for 40 min at 4 °C. The resulting supernatant was collected and centrifuged again at $100,000 \times g$ for 70 min. The exosome pellet was resuspended in PBS for RNA extraction using TRIzol. The isolation of exosomes from mouse plasma was performed following the user manual of the kit (Invent Biotechnologies, USA). Human plasma exosomes were collected following the instructions of ExoQuick-TC Exosome Precipitation Solution Reagent (EXOQ5TM-1; System Biosciences, Palo Alto, CA, USA). Briefly, about 250 μl plasma sample was centrifuged at $3000 \times g$ for 15 min, and then the supernatants were blended by adding four times the volume of ExoQuick Exosome Precipitation Solution, followed by incubation at 4 °C for 30 min. The mixture was centrifuged at $1500 \times g$ for 30 min to pellet exosomes. The exosomes were added to a copper electron microscopy grid and stained with phosphotungstic acid for 30 s. Grids were air-dried and exosome micrographs were captured using a transmission electron microscope (Leica Microsystems, Wetzlar, Germany) at 120 kV. Nanoparticle tracking analysis was performed using NanoSight NS300 (Malvern Panalytical, Malvern, UK) to observe the concentration and size distribution of the exosomes. Furthermore, the characterization of the exosomes was confirmed by the presence of exosomal protein markers by western blot using the following antibodies: anti-CD9 (1:1000, ab92726; Abcam), anti-CD63 (1:2000, ab217345; Abcam), anti-CD81 (1:1000, #10037, Cell Signaling Technology, Danvers, USA), and anti-Tsg101 (1:1000, ab125011; Abcam).

## Fluorescence in situ hybridization

To detect the expression and visualize the localization of circRNA in intestinal tissue sections, we performed FISH with Cy3-labeled probe sequences specific to the head-to-tail conserved junction of circEZH2_005 (RiboBio, Guangzhou, China). The tissue slides were deparaffinized and permeabilized in PBS containing 0.3% Triton X-100 and then incubated with proteinase K. The samples were incubated with the probe overnight at 37 °C in a moist chamber. The samples were washed with 4×saline sodium citrate (SSC) at 42 °C for 20 min, with 2×SSC for 10 min, and twice with 1×SSC for 10 min. Furthermore, RNA-FISH was performed using a FISH Kit (RiboBio). Images were obtained at 200× magnification using an immunofluorescence microscope (Olympus).

## CircEZH2_005 pulldown assay

The biotin-labeled circEZH2_005 and oligonucleotide probes (Ribo-Bio) were mixed with streptavidin magnetic beads (Beaver, Suzhou, China) in RIP buffer for 4 h. Subsequently, the cell lysate was incubated with the probe complex for 12 h at 4 °C. After purification, the enriched circEZH2_005 was quantified through qRT-PCR. Furthermore, the bound circEZH2_005 proteins were identified using western blot and mass spectrometry analyses. The mutant circEZH2_005 RNA fragments were transcribed with T7 RNA polymerase and labeled with

biotin according to the manufacturer's instructions (GeneChem). Different hnRNPA1 constructs were obtained by purification (constructed into plasmids and 3xFlag).

## RNase R and actinomycin D treatment

Total RNA (2 μg) was incubated with or without 3 U/μg of RNase R (Epicentre Technologies, Madison, WI, USA) for 15 min at 37 °C. The expression of circEZH2_005 and other RNAs was detected through qRT-PCR. Cells were seeded in six-well plates.Twenty-four hours later, cells were exposed to 2 μg/ml actinomycin D or dimethyl sulfoxide (Sigma-Aldrich, St. Louis, USA) and collected at indicated time points. The RNA stability of Gprc5a was analyzed using qRT-PCR.

## Organoid culture and transfection

We isolated Lgr5+ ISCs from Lgr5-GFP mice and used them to generate crypt-villus organoids. Organoids were treated with 1×TrypLE Express (Thermo Fisher Scientific) for single-cell preparations. Cell transfection and co-transfection experiments were performed using Lipofectamine 3000 (Thermo Fisher Scientific) according to the manufacturer's instructions. After transfection for 48 h, cells were challenged with H/R, and different assays were conducted. siRNAs targeting circEZH2_005 (RiboBio), hnRNPA1, and Gprc5a (HippoBio, Zhejiang, China) were used and lentivirus-mediated short hairpin RNA (shRNA) targeting circEZH2_005 and control lentiviral vectors (Obio, Shanghai, China) were used. All siRNA sequences are listed in Supplementary Table S4. circEZH2_005-GFP-expressing adenovirus upregulated (GeneChem). To explore the protective effect of circEZH2_005 on intestinal I/R in vitro, the organoids were randomly divided into four groups: (1) Control + Sh-NC, Control + Sh-circEZH2_005, H/R + Sh-NC, H/R + Sh-circEZH2_005. (2) Control + Vector, Control + circEZH2_005, H/R + Vector, H/R + circEZH2_005. The number of all organoid experiments was $n = 3$. The investigator who established the intestinal H/R model was blinded to the experimental design and group allocation.

## Enzyme-linked immunosorbent assay (ELISA)

Human fatty acid binding protein-2 (IFABP) protein levels (DFBP20; R&D Systems, Minneapolis, USA), human D-lactate (ab83429; Abcam), murine TNF-α (EK282/4-96; MultiSciences, Hangzhou, China), and murine IL-β (KGEMC004; KeyGEN Biotech, Nanjing, China) in plasma were evaluated using commercially available ELISA kits according to the manufacturer's instructions.

## Statistical analysis

All statistical analyses were performed using GraphPad Prism 8 software (GraphPad Software, Inc., La Jolla, CA, USA). Quantitative data are presented as the means ± standard deviations. For all animal and cell experiments with a small sample size ($n ≤ 6$), the Shapiro−Wilk test was used to test for normality with a threshold of 0.05. For all clinical samples, the D'Agostino−Pearson test was used to test for normality with a threshold of 0.05. For data with normal distribution, the Student's $t$-test was used to analyze the significance of differences between two means; to compare several groups, one-way ANOVA with Tukey's or Dunnett's multiple-comparison test was used; correlation analysis was performed using Pearson correlation. For non-normally distributed data, the Mann−Whitney or Kruskal−Wallis test was used to test the significance of differences, and correlation analysis was performed using Spearman correlation. Receiver operating characteristic curves were constructed to assess the diagnostic accuracy of exosomal circRNAs, and logistic regressions were calculated to generate a diagnostic value of the combination of exosomal circRNAs and IFABP data. Results with $P < 0.05$ were considered significant.

## Reporting summary

Further information on research design is available in the Nature Portfolio Reporting Summary linked to this article.

## Data availability

All data needed to evaluate the conclusions of the paper are presented in the paper and/or the Supplementary Materials. RNA-sequencing data that supported the findings of this study have been deposited in GEO G SE227355, and G SE227518. The mass spectrometry proteomics data have been deposited to the ProteomeXchange Consortium (http://proteomecentral.proteomexchange.org) via the iProX partner repository[57,58] with the dataset identifier PXD040925. Source data areprovided with this paper. Source data are provided with this paper.

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

## Acknowledgements

We thank Pro. Peng Chen for helpful discussions and would like to thank Editage (www.editage.cn) for English language editing. This work was supported by grants from National Natural Science Foundation, Beijing, China (82102296 to Wenjuan Zhang), National Natural Science Foundation, Beijing, China (81671955 to Kexuan Liu), Key Program of National Natural Science Foundation, Beijing, China (81730058 to Kexuan Liu), and Basic and Applied Basic Research Project, Guangzhou, China (202201011065 to Wenjuan Zhang).

## Author contributions

W.J.Z., B.W.Z., and K.X.L. conceived and designed the project. C.L. and B.W.Z. provided administrative support and revision suggestion. W.J.Z., C.L., and B.C.Z. performed all clinical experiments. W.J.Z. and B.W.Z. performed most of the biochemical and molecular experiments, with the assistance of X.Y., J.Z. and F.L.Z.. F.Z., J.C., and Z.W.Y. performed all animal experiments and analyzed all animal data. Y.Q.D., S.T.Z., Y.F.Y., Q.S.S., and Z.B.L. collected clinical samples and analyzed data, performed PCR, histology and ELISA experiments. W.J.Z., B.W.Z., and K.X.L. wrote the paper with the assistance of J.J.H., F.D., and W.F.L.. All authors read and approved the final manuscript.

## Competing interests

The authors declare no competing interests.
