## [Peer Review File · Nature Communications]

Exosomal circEZH2_005, an intestinal injury biomarker, alleviates intestinal ischemia/reperfusion injury by mediating Gprc5a signalingREVIEWER COMMENTS

Reviewer #1 (Remarks to the Author):

Summary:

This is a novel study that attempts to define a marker for I/R injury and finds a circular RNA molecule that is downregulated in high-grade injuries involving I/R but not in low grade. The authors provide strong evidence that this circular RNA is stably expressed in exosomes and that it can regulate the intestinal stem cell function. The authors attempt to define the cellular function of circEZH2_005 and find that it interacts directly with hnRNPA1 and that interaction facilitates the expression of Gprc5, a factor induced by hypoxic signaling.

Major comments:

- 1) Is the I/R injury the only source of circEZH2_005? Would these circular RNAs be released during other intestinal insults, potentially confounding the value of the finding. For example, authors need to address if IBD and viral/bacterial infections of the intestine would produce similar serum profile of circEZH2_005, which needs to be characterized in order to use this marker in I/R injuries.
- 2) The researchers localize to the jejunal crypts, but the significance of this localization is left unexplored, as to why this are releases the exosomes. What are the expression patters in other areas of the intestine and is the large intestine also involved in the production of this circular RNA?
- 3) Authors do not specify which kit is used to purify the human exosomes from patients and how rapidly this procedure is, which is of significance in clinical diagnosis and evaluation of suitability of this marker for I/R diagnosis. How long would a purification take if it was to be used to diagnose a patient?
- 4) In figure 5-7 where biochemical experiments are performed by overexpression of circEZH2_005, the authors should use a mutated version of the circular RNA with the mutation in the hnRNPA1 which they identify as a direct binder to the circEZH2_005. In the crypt rescue experiments involving overexpression of circEZH2_005, using a mutated version of circEZH2 expressed from a virus to assess the ability increase the number of stem cells would be necessary.
- 5) The authors should test whether the decrease /induction of circEZH2_005 in the intestine

is regulated by hypoxia inducible factors (HIFs). With a large number of reports showing induction of circular RNA by HIFs in cancer and other disease, it begs a question if these factors are involved in regulating stem cell activity during I/R, and it has clinical significance with a number of clinically relevant drugs that regulate HIF activity. Mice with conditional deletion of HIFs in the intestinal stem cells need to be used to address this question.

Minor comments:

- 1) In figure 7 the micrographs of the intestine are provided but no histological quantitation is included.
- 2) The description of how Lrg5+ stem cells were purified is missing in the methods.
- 3) The explanation for Chiu's score and its significance needs to be explained. Similarly, the significance of findings in figure 3H needs to be explained.
- 4) In lines 190-193 of the manuscript references and a context need to be provided for circRNAs involvement in disease regulation.
- 5) In the methods section when researchers are blinded to the samples, it needs to be stated clearly and grammatically.

Reviewer #2 (Remarks to the Author):

major concerns

1. The authors first measured the levels of CircEZH2_005 (EZH2 for short) in the plasma of patients undergoing cardiopulmonary bypass at various postoperative time points. It is true that postoperative cardiopulmonary bypass can cause inadequate blood supply to the small intestine, but the severity of ischemia is much lower than in the animal model applied in this paper. How to explain this mismatch.
2. The mouse IIR model uses the ischemia 60 min and reperfusion 120 min timepoint. Although a minority of articles also use this time-point, reading through most articles on intestinal ischemia-reperfusion injury in mice, it is rarely used by investigators because very few mice can successfully survive to 120 minutes of reperfusion (even if the ischemic time is 30-45 minutes). Furthermore, 60 minutes of SMA ischemia in mice will cause widespread

intestinal necrosis (not applicable to the study of various programmed cell deaths), such that mice will have difficulty surviving even through the 120-minute reperfusion period. This differs too much from the intestinal ischemia in patients undergoing clinical cardiopulmonary bypass. I wish the authors could provide a more comprehensive experimental procedure, an atlas, and operation specification for 60 minutes of intestinal ischemia and 120 minutes of reperfusion in mice, including whether/how to use anesthesia plan and ventilator, detailed anesthesia protocol, surgical procedures, the model of vascular clamps, perioperative monitoring methods, and fluid rehydration methods.

3. IFABP is a classical biomarker of intestinal ischemia-reperfusion injury. If EZH2 and IFABP are "similar", what is its advantage? Does it have advantages in terms of sensitivity, specificity, etc.?

4. Statistical problems: What is the reason for using standard errors instead of standard deviations? Have all data been tested for normal distribution and variance homogeneity? How are some data that apparently fail to be normally distributed statistically different?

minor concerns

1. In Figure 5a, HR should be IR.

2. Note the font of some of the Greek letters.

Reviewer #3 (Remarks to the Author):

The study is well-structured with novel findings. It is worth publication; however, I suggest several amendments prior to it can be accepted. Methods are explained in detail, and the figures possess good quality.

The quality of the writing is not satisfying and requires vigorous editing and improvement. The introduction is weak, unnecessarily long, and lacks a dynamic flow. circRNAs are not

necessarily carried by exosomes. The authors are suggested to briefly discuss the structure, function, and clinical significance of exosomal circRNAs in a dynamic flow. A review of recent studies highlighting the importance of exosomal circRNAs in I/R injury and the novelty of the current study should be added appropriately to the introduction.

2- 50 patients to evaluate exosomal circRNA expression seems a small population

3-Investigation of circEZH2_005 distribution in intestinal tissues seems unnecessary for discussing in a title

4-All claims should be treated cautiously using some terms like suggesting, may, can, etc.

5- The discussion is too long and requires shortening with a focus on explaining the study findings and their significance in therapy and diagnosis

Response Letter

Thank you for the reviewers' professional comments concerning our manuscript entitled “Exosomal circEZH2_005, an intestinal injury biomarker, alleviates intestinal ischemia/reperfusion injury by mediating hnRNPA1/Gprc5a signaling”(ID: NCOMMS-22-53087). We are very grateful to the reviewers for taking the time and effort to review the manuscript so carefully. I really appreciate all your comments and suggestions! Those comments are very valuable and helpful in improving the quality of our paper. We have made extensive modifications to our manuscript and supplemented extra data to make our results convincing according to the reviewers' comments. Revised portions are marked in red in the revised version. Meanwhile, the manuscript language has been edited by the language service of Editage (www.editage.cn). The main corrections in the paper and the responses to the reviewer's comments are as follows:

Responds to the reviewers' comments:

Reviewer #1:

Major comments:

1. Response to comments: (Is the I/R injury the only source of circEZH2_005? Would these circular RNAs be released during other intestinal insults, potentially confounding the value of the finding. For example, authors need to address if IBD and viral/bacterial infections of the intestine would produce similar serum profile of circEZH2_005, which needs to be characterized in order to use this marker in I/R injuries.)

Response: We appreciate the reviewer's insightful suggestion and we recognize that the diagnostic specificity of circEZH2_005 should be further validated. In IBD model mice, we found that circEZH2_005 expression was elevated in intestinal tissue compared with the control group. However, no significant differences in circEZH2_005 expression were observed between the two groups in plasma exosome (**Figure A**). In CLP mice as bacterial infections model, circEZH2_005 expression showed no statistical difference compared to sham-operated group both in the intestinal tissues and plasma exosome, although the expression of circEZH2_005 in plasma exosomes showed a tendency to increase (**Figure B**). From the above results, we believe that the specificity of circEZH2_005 as a marker of intestinal I/R injury is acceptable. But it needs to be validated in more disease models. The

experimental results are shown in the figure below. We thank again for the profound insights provided by the reviewer.

2. Response to comments: (The researchers localize to the jejunal crypts, but the significance of this localization is left unexplored, as to why this are releases the exosomes. What are the expression patters in other areas of the intestine and is the large intestine also involved in the production of this circular RNA?)

Response: We thank the reviewer for raising this important point. Our group has found that in the jejunal of mice, circEZH2_005 was expressed in the epithelium, while predominantly located in jejunal crypt. Crypts are subunits that house intestinal stem cells (ISCs) and have been recognized as responsible for intestinal regeneration a vital process for the homeostatic self-renewal and the response to all kinds of stress injuries, including irradiation, exposure to food toxins and ischemia/reperfusion (PMID: 27802133). Notably, our preliminary studies showed that there was also circEZH2_005 expression in ISCs. ISCs may be an effective therapeutic target to promote intestinal recovery from intestinal ischemia/reperfusion injury (PMID: 30705108). In addition to the basic ability of regeneration and re-placement, ISCs also act by paracrine actions. Under certain conditions, a variety of paracrine factors can be secreted by stem cells, which exert different biological effects such as anti-inflammation, anti-apoptosis, and promoting proliferation. A large body of literature has shown that stem cells can play a role in the treatment of diseases including necrotizing enterocolitis by

secreting exosomes (PMID: 34090496). These studies suggest that ISCs in the jejunal crypts can promote intestinal tissue damage repair by secreting exosomes. Therefore, we hypothesized that circEZH2_005 might be released into the blood by the jejunal crypts cells via exosomes, which further indicated that circEZH2_005 has tissue-specific and can be served as a indicator for intestinal injury.

To explore whether the large intestine is involved in circEZH2_005 release, the localization and expression of circEZH2_005 in the large intestine were detected in I/R model mice. FISH and qPCR results showed that circEZH2_005 was not differentially expressed in I/R group compared with the sham group (**Figure A and B**). FISH also showed that the expression of circEZH2_005 in large intestine was not predominantly expressed in stem cells (**Figure B**). Together, these data suggest that the large intestine is not involved in the differential expression patterns of circEZH2_005 during intestinal I/R.

3. Response to comments: (Authors do not specify which kit is used to purify the human exosomes from patients and how rapidly this procedure is, which is of significance in clinical diagnosis and evaluation of suitability of this marker for I/R diagnosis. How long would a purification take if it was to be used to diagnose a patient?)

Response: Thank you for your comments. We apologize for this; we have added the human plasma exosome extraction kit and procedure [Human plasma exosomes were collected following the instructions of ExoQuick-TC Exosome Precipitation Solution Reagent (EXOQ5™-1; System Biosciences,

Palo Alto, USA). Briefly, about 250µl plasma sample was centrifuged at 3000 g for 15 minutes and then the supernatants were blended by addition of four times volume of ExoQuick Exosome Precipitation Solution, followed by incubation at 4°C for 30 minutes. The mixture was centrifuged at 1500 g for 30 minutes to pellet exosomes.] to the section of Materials and Methods (page 17. line 538-545 of the revised manuscript). The extraction time of plasma exosomes was about 2 hours. Extraction of exosomes and RNA and subsequent qPCR can be completed within 4 hours.

4. Response to comments: (In figure 5-7 where biochemical experiments are performed by overexpression of circEZH2_005, the authors should use a mutated version of the circular RNA with the mutation in the hnRNPA1 which they identify as a direct binder to the circEZH2_005. In the crypt rescue experiments involving overexpression of circEZH2_005, using a mutated version of circEZH2 expressed from a virus to assess the ability increase the number of stem cells would be necessary.)

Response: We are deeply thankful to the reviewer for raising this important point as we realized that the data we presented was not sufficient, and we have done it according to your ideas. We constructed deletion mutant lentiviral of the hnRNPA1 with the truncated RRM2 region and circEZH2 mutated version lentiviral lacking 178-223 base sequences respectively, and then designed functional rescue experiments. As expected, compared with the vector group, co-transfection of wild-type circEZH2_005 and wild-type hnRNPA1 virus significantly alleviated the morphological injury of the organoid during H/R, while the mutation of either circEZH2_005 or hnRNPA1 significantly reversed the protective effect (Supplementary Figure 9I in the revised manuscript). In addition, since the hnRNPA1 virus carries a Flag tag, we detected the localization of hnRNPA1 with Flag antibody. As shown, the hnRNPA1 protein exhibited both nuclear and cytoplasmic localization in cell co-transfection of wild-type circEZH2_005 and wild-type hnRNPA1 virus. However, the mutation of either circEZH2_005 or hnRNPA1 showed mainly nucleus fluorescence (Supplementary Figure 9G in the revised manuscript). Moreover, the mutated version of circEZH2_005 or the mutation in the hnRNPA1 prevented the upregulation of Gprc5a protein levels upon H/R (Supplementary Figure 9H in the revised manuscript). Please see the Results section, lines 258 - 264, page 8 of the revised paper.

In the crypt rescue experiments involving overexpression of circEZH2_005, we overexpressed circEZH2_005 lentiviral lacking the hnRNPA1-binding region (circEZH2_005 MUT) in ISCs. The capacity of organoid survive (Supplementary Figure 8G in the revised manuscript) and proliferation (Supplementary Figure 8H in the revised manuscript) was almost abrogated in circEZH2_005 MUT overexpressed cells, suggesting that the interaction of circEZH2_005 with hnRNPA1 is necessary for the intestinal crypt cell proliferation during H/R injury. For the text description, please see the Results section, lines 224 - 229, page 7 of the revised paper. Thanks again to the reviewer for improving the quality of our articles.

5. Response to comments: (The authors should test whether the decrease /induction of circEZH2_005 in the intestine is regulated by hypoxia inducible factors (HIFs). With a large number of reports showing induction of circular RNA by HIFs in cancer and other disease, it begs a question if these factors are involved in regulating stem cell activity during I/R, and it has clinical significance with a number of clinically relevant drugs that regulate HIF activity. Mice with conditional deletion of HIFs in the intestinal stem cells need to be used to address this question.)

Response: We appreciate it very much for this precious comments and suggestions. We agree with you that if HIFs induce circEZH2_005 expression, it is of great significance to regulate stem cell activity during I/R through the application clinical drug of HIF. HIF is a heterodimer consisting mainly of α and β subunits. HIF- β is stably expressed in cells and plays a structural role. Hif- α is regulated by a hypoxia signal and is the active subunit of HIF. Studies have shown that HIF- α and HIF-2 α are transcription factors in response to changes in oxygen concentration under hypoxic conditions. Therefore, we examined the effects of HIF-1 α and HIF-2 α on circEZH2_005 expression. qPCR results showed that the expression of circ-EZH2_005 was not significantly changed after interference of HIF-1 α , although HIF-1 α depletion decreased the expression of EZH2 gene (**Figure A**). Since we found the expression of HIF-2 α in intestinal epithelial cells is so low as to affect its interference efficiency (**Figure B**), we did not consider it necessary to verify whether it affects circEZH2_005 expression. These results preliminarily indicated that circEZH2_005 was not regulated by HIF, although HIF-1 α could regulate EZH2 expression.

Circular RNAs are covalently closed structure with neither 5' to 3' polarity nor polyadenylated tail. Studies have uncovered that back-splicing requires canonical spliceosomal machinery and can be facilitated by both complementary sequences (cis-acting elements) and specific protein factors (trans-acting factors). In addition, the expression of circRNA is also affected by the expression of its parent genes, nucleoplasm transport and degradation (PMID: 31395983). Our results show that the expression of circEZH2_005 is not completely regulated by HIF- α and EZH2 gene, and we suspect that it may also be influenced by splice factors under hypoxia. Unfortunately, due to the influence of time, we have not been able to further elucidate the upstream factors that regulate circ-EZH2_005 expression. Also, this is not the focus of this study. We will further explore the specific upstream regulatory mechanism of circEZH2 and the key factors regulating stem cell activity during I/R in future work. We are still very grateful to the reviewer for the valuable comments, which made us realize more clinical significance of HIFs in regulating stem cell activity. Thanks again to the reviewer.

Minor comments:

1. Response to comments: (In figure 7 the micrographs of the intestine are provided but no histological quantitation is included.)

Response: Thank you for your comments. We apologize for it. We have supplemented the histological quantitation of Figure 7. Please see Figure 7N in the revised version.

2. Response to comments: (The description of how Lrg5+ stem cells were purified is missing in the methods.)

Response: Thank you for your comments. We apologize for the absence of clearly describing the purification of Lrg5+ stem cells. We have supplemented the whole process separately in the method. Please see page 15, lines 460 – 476 in the Materials and Methods section of the revised paper.

3. Response to comments: (The explanation for Chiu’ s score and its significance needs to be explained. Similarly, the significance of findings in figure 3H needs to be explained.)

Response: Thank you for your comments. We are very sorry for our negligence in the explanation for Chiu’s score. We have supplemented the explanation for Chiu’s score separately in the method. Please see page 14, lines 447 – 458 in the Materials and Methods section of the revised paper.

For the significance of findings in figure 3H, first of all, circRNA has been shown to be tissue and cell-specific, its subcellular localization has important biological implications. To study the functional mechanism of circRNA, it is necessary to know the localization of circRNA in cells. The corresponding functional mechanism of circRNA varies with different localization of circRNA. CircRNA localized in the nucleus may be involved in gene transcriptional regulation, while circRNA in the cytoplasm is mainly involved in gene regulation at the post-transcriptional level by binding miRNAs or proteins. Our results show that circEZH2_005 is mainly localized in the cytoplasm, which provides a basis for the

subsequent study of the interaction protein mechanism. On the other hand, FISH results showed that circEZH2_005 localized in the cytoplasm in mouse intestinal tissue, which needed to be further confirmed by nuclear plasma separation experiments to determine whether circRNA could be secreted into exosomes. Thanks again for the comments of the reviewer.

4. Response to comments: (In lines 190-193 of the manuscript references and a context need to be provided for circRNAs involvement in disease regulation.)

Response: Thank you very much for discovering this error. As suggested by the reviewer, we have added more references to support this idea. Please see page 6, lines 193-194 in the Results section of the revised paper.

5. Response to comments: (In the methods section when researchers are blinded to the samples, it needs to be stated clearly and grammatically.)

Response: Thank you for your comments. We are very sorry for our incorrect writing and it has been rectified. Please see lines 497 - 499 of page 16 and lines 596 - 597 of page 19 in the revised manuscript.

Reviewer #2:

1. Response to comments: (The authors first measured the levels of CircEZH2_005 (EZH2 for short) in the plasma of patients undergoing cardiopulmonary bypass at various postoperative time points. It is true that postoperative cardiopulmonary bypass can cause inadequate blood supply to the small intestine, but the severity of ischemia is much lower than in the animal model applied in this paper. How to explain this mismatch.)

Response: We thank the reviewer for pointing this out. Your question is very reasonable. We agree with you that the intestinal I/R animal model we use is different from the situation of intestinal injury caused by clinical cardiopulmonary bypass. This is also one of a limitation of our research, and we recognize this limitation should be faced up and discussed. We know that an important principle in designing animal disease models is that the replicated model should be as close as possible to the situation of human diseases. However, currently, many animal models cannot fully replicate the true situation of human diseases, including intestinal I/R models, sepsis models, IBD models, etc. Therefore, the animal model is only an indirect study, and may only be similar to human disease in a local or several aspects. Conclusions from animal models must eventually be verified in humans.

In general, mice have a much higher tolerance to intestinal ischemia than humans. After clamping the superior mesenteric artery for 60 minutes, about 50% of the mice survived for more than 24 hours. But things may be much different in humans, which is similar to the myocardial ischemia-reperfusion model. In addition, in some models that require gavage, the dosage given to animals is often greater than that of humans. Moreover, due to the high tolerance of mice, if the ischemia-reperfusion time is too short, it is often not possible to obtain damaged intestinal tissue samples. Therefore, we chose the ischemia-reperfusion time that is prone to damage for animal model construction.

To screen the marker that can predict intestinal injury in time, we designed multiple time points (45min, 45min/1h, 45min/2h, 45min/4h, 1h, 1h/1h, 1h/2h, 1h/4h) to verify the expression levels of circRNA, representing different degrees of injury. We found that circEZH2_005 expression was also down-regulated at 45min ischemia, but the most significant differences were 1h/1h and 1h/2h, which means circEZH2_005 may also be validated in clinical cases of mild intestinal injury. Finally, in the previously published paper by our research group, an intestinal I/R animal model with ischemia for 1 hour and reperfusion for 2 hours was used, and it was validated in clinical patients with an intestinal injury during cardiopulmonary bypass (PMID: 35241180, 33779497). Therefore, we believe that the detection of circEZH2_005 levels in the plasma of CPB patients can verify the conclusions of our animal model to a certain extent, but more samples are needed for further confirmation. Thank you again for the profound insights provided by the reviewer.

2. Response to comments: (The mouse IIR model uses the ischemia 60 min and reperfusion 120 min timepoint. Although a minority of articles also use this time-point, reading through most articles on intestinal ischemia-reperfusion injury in mice, it is rarely used by investigators because very few mice can successfully survive to 120 minutes of reperfusion (even if the ischemic time is 30-45 minutes). Furthermore, 60 minutes of SMA ischemia in mice will cause widespread intestinal necrosis (not applicable to the study of various programmed cell deaths), such that mice will have difficulty surviving even through the 120-minute reperfusion period. This differs too much from the intestinal ischemia in patients undergoing clinical cardiopulmonary bypass. I wish the authors could provide a more comprehensive experimental procedure, an atlas, and operation specification for 60 minutes of intestinal ischemia and 120 minutes of reperfusion in mice, including whether/how to use anesthesia plan and ventilator, detailed anesthesia protocol, surgical procedures, the model of vascular clamps, perioperative monitoring methods, and fluid rehydration methods.)

Response: We appreciate the reviewer's insightful suggestion and agree that our animal I/R model of 60 minutes of SMA ischemia and 120 minutes of reperfusion is different from the human. In fact, the usual time point for intestinal I/R modeling in our laboratory was 1h of

ischemia and 2h of reperfusion (PMID: 36229053, 35241180, 33779497). We also believe that a specific description of the animal experiment process is very necessary. We apologize for the absence of a clear and detailed animal experimental procedure. As suggested by the reviewer, we have revised the text in the Materials and Methods section of the revised paper, and hope that it is now clearer. Please see page 13, lines 418 - 446 of the revised manuscript. For the corresponding process atlas, please see below. Thank you again for the valuable insights which made our article more rigorous.

workflow of mouse intestinal IR injury model

3. Response to comments: (IFABP is a classical biomarker of intestinal ischemia-reperfusion injury. If EZH2 and IFABP are "similar", what is its advantage? Does it have advantages in terms of sensitivity, specificity, etc.?)

Response: We thank the reviewer for raising this important point. A vast array of laboratory biomarkers has been evaluated in the diagnosis of acute intestinal ischemia, but an ideal biomarker (rapid, stable, highly specific and sensitive, inexpensive and easy to be measured) is still seemingly missing. Although IFABP is a relatively recognized marker for the

diagnosis of intestinal injury, it still has many limitations (PMID: 31555708, 28395790). Studies have shown that although IFABP alone has a high sensitivity in the diagnosis of certain diseases (such as strangulated ileus or necrotizing enterocolitis of newborns), it lacks specificity (PMID: 19424744). So, at present, IFABP is not perfect enough to be used solely. And studying the combination outcome of IFABP and circEZH2_005 is probably a better way. Although the AUC value of circEZH2_005 to distinguish intestinal I/R injury was relatively low (0.732), it increased the diagnostic efficacy of the common enterocyte injury marker IFABP from an AUC of 0.776 to 0.846, reflecting a good clinical application value.

CircRNAs are becoming one of the most promising markers of human disease due to their terminal closed-loop structure being more stable than linear RNA, their long half-life, and their high specificity. More importantly, the tissue specificity of circRNAs is higher than that of the mRNA. These characteristics suggest that circRNAs may have better analytical effectiveness, including analytical specificity, accuracy, reproducibility and repeatability, when used as biomarker molecules compared to other molecules. The sensitivity and specificity of circEZH2_005 and IFABP were calculated by ROC curve, and the results showed that the sensitivity and specificity of circEZH2_005 were 51.28 and 90.91%, while the sensitivity and specificity of IFABP were 74.36 and 72.73%, respectively. Indicating that circEZH2_005 has an advantage over IFABP in specificity. However, the specificity of circEZH2_005 needs to be further verified in more other intestinal diseases.

With the development of circRNA as a marker of disease diagnosis, the time required for circRNA detection is shorten. For instance, it has been recently reported that nucleic acid can be detected rapidly, sensitively, and specifically in clinical samples using the loop-mediated isothermal amplification assay or CRISPR diagnostics (PMID: 33288960). Therefore, in clinical settings, the detection of circEZH2_005 level in a patient's blood would be a promising approach for a fast, noninvasive, cheap, and simple method for the detection of patients with intestinal ischemia-reperfusion injury. We thank the reviewer again for providing such profound insights.

4. Response to comments: (Statistical problems: What is the reason for using standard errors instead of standard deviations? Have all data been tested for normal distribution and variance homogeneity? How are some data that apparently fail to be normally distributed statistically different?)

Response: We are very grateful to the reviewer for pointing out our problems in statistics. We are very sorry that the knowledge of statistics is our weakness. We have consulted the statistical experts and believe that SD is more suitable for our statistical analysis of results than SEM, so our data are presented as mean \pm SD in the revised manuscript. In addition, our data were not been tested for normal distribution and variance homogeneity, we are very

sorry and ashamed of this. According to the suggestions of reviewers, we have carried out normal distribution and variance homogeneity tests for all data in the revised manuscript. All normal distribution and variance homogeneity test results are shown in the response table (**Response table for normal distribution test**). We found that the data failure to be normally distributed would result in the change in p value after use of the nonparametric test, but it did not change the statistical significance of our data.

In addition, we have re-analyzed all the data according to the analysis methods recommended by statistics experts. Statistical analysis showed similar result to the previous analysis, which did not affect the conclusion of our study. Please see page 19, lines 605 - 620 in the Materials and Methods section of the revised paper for the detailed statistical analysis methods. We appreciate for reviewers' warm work earnestly and hope that the statistical analysis method can be corrected in time and obtain your forgiveness. We apologize again for the errors in statistics, and we are also very grateful for the guidance of the reviewer, which will enable us to conduct rigorous and scientific processing of data in future scientific research. We hope to learn more from you.

5. Response to comments: (In Figure 5a, HR should be IR.)

Response: We thank the reviewer very much for your reminder. We are very sorry for our careless mistake and it has been rectified.

6. Response to comments: (Note the font of some of the Greek letters.)

Response: Thanks for your careful checks. We feel sorry for our carelessness. In our resubmitted manuscript, the error is revised. Thanks for your correction.

Reviewer #3:

1. Response to comments: (The quality of the writing is not satisfying and requires vigorous editing and improvement.)

Response: We thank the reviewer for underlining this deficiency. We apologize for the poor language of our manuscript. We worked on the manuscript for a long time and the repeated addition and removal of sentences and sections obviously led to poor readability. We have now worked on both language and readability and have also carefully revised the writing with a professional language editing service to improve the grammar and readability. We hope that the flow and language level have been substantially improved. Thanks again to the reviewer for the valuable comments.

2. Response to comments: (The introduction is weak, unnecessarily long, and lacks a dynamic flow. circRNAs are not necessarily carried by exosomes. The authors are suggested

to briefly discuss the structure, function, and clinical significance of exosomal circRNAs in a dynamic flow. A review of recent studies highlighting the importance of exosomal circRNAs in I/R injury and the novelty of the current study should be added appropriately to the introduction.)

Response: We thank the reviewer for the suggestion. We regret that there are problems in the writing of the introduction. This section was revised and modified according to the information suggested by the reviewer to make the expression more clear and accurate. We hope the revised manuscript could be acceptable for you. Please see the introduction section of the revised paper.

3. Response to comments: (2- 50 patients to evaluate exosomal circRNA expression seems a small population.)

Response: We thank the reviewer for underlining this deficiency. The sample size of our study was calculated from the estimated area under the curve and the width of the confidence interval (PMID: 34851395). Our prospective sample size calculation method: Assuming an AGI incidence of 75% and a discrimination magnitude of approximately 0.70 and a confidence interval of ± 0.19 (to obtain a statistically significant result), then a total of 35 samples with which 26 were positive and 9 were negative were required.

The primary aim of our results is to evaluate whether circEZH2_005 can distinguish patients with intestinal I/R injury from those without. So we only used 50 samples for evaluation. But we agree with you that 50 clinical specimens are very small to estimate the diagnostic potential of circEZH2_005. These results need to be validated in a larger multicenter cohort to evaluate the diagnostic performance of circEZH2_005 before its translation into routine clinical practice. This is also the work that we need to further research in the future. We thank the reviewer again for raising this important point.

4. Response to comments: (3-Investigation of circEZH2_005 distribution in intestinal tissues seems unnecessary for discussing in a titl.)

Response: Thank you for the title suggested. We have changed [Characterization and distribution of circEZH2_005 in intestinal tissues] in the precedent version to [Characterization and expression of circEZH2_005] in the revised paper. Please see page 5, line 142 of the Results section.

5. Response to comments: (4-All claims should be treated cautiously using some terms like suggesting, may, can, etc.)

Response: We sincerely thank the reviewer for careful reading. As suggested by the reviewer, we have modified some inappropriate expressions throughout the text according to the comment. For example, we have changed [we proposed the hypothesis that detecting changes in intestinal exosome content after intestinal I/R could help identify specific and sensitive markers of intestinal injury] in the discussion section to [we hypothesized that detecting changes in intestinal exosome content after intestinal I/R might help identify an ideal marker of intestinal injury] in the revised paper. We have changed [Consequently, our data suggest that circEZH2_005 is a useful therapeutic target, and

circulating exosomal circEZH2_005 may serve as a biomarker for the early diagnosis of intestinal I/R injury] in the discussion section to [Consequently, our data raise the possibility that circEZH2_005 may be a useful therapeutic target for rescuing the ischemic intestinal tissue in I/R injury.] in the revised paper. Phrases like “novel”, “new”, “for the first time” have been avoided throughout the manuscript. We hope that the correct expression is presented after correction. Thanks again to the reviewer.

6. Response to comments: (5- The discussion is too long and requires shortening with a focus on explaining the study findings and their significance in therapy and diagnosis.)

Response: We thank the reviewer for pointing this out. We apologize for the lengthy and inappropriate writing in the discussion section. We have tried our best to polish the language and content of the discussion section according to the opinions of the reviewer. We hope the current writing of the revised paper can be concise and highlight the key points. Thanks again to the reviewer for improving the quality of our articles.

We sincerely thank all reviewers again for your valuable feedback that we have used to improve the quality of our manuscript. All your suggestions are very important and have important guiding significance for our future scientific research work. Thanks to your suggestions, We have found our shortcomings in our current work. If there are any other modifications we could make, we would like very much to modify them and we really appreciate your help.

REVIEWERS' COMMENTS

Reviewer #1 (Remarks to the Author):

Remarks to the Author:

The authors have made a commendable effort in addressing all major and minor points that were detailed in the initial review. Portions of the manuscript were revised accordingly and important controls are now included in the biochemical analysis of the circEZH2_005 and hnRNPA1. The initial conclusions made by the authors appear to stand unchanged. The critical and minor points were addressed in a satisfactory manner and the manuscript has been greatly improved by careful revision of writing style and re-working the statistics. Before publication I ask that the authors consider adding evidence that the intestine subjected to the lengthy ischemic insult followed by reperfusion is not highly necrotic at the starting point their analysis.

Reviewer #2 (Remarks to the Author):

I believe the author has addressed my question directly. Although there may still be room for discussion and improvement on some aspects, it provides a solid answer that is quite satisfactory in the current research field. This paper should be considered for acceptance.

Reviewer #3 (Remarks to the Author):

Many thanks for revising the manuscript and responding to comments. Amendments are properly replied to and applied within the revised manuscript. Now, I have no further comments and can endorse its publication.

Best,

Sajad Najafi, Ph.D

We would also like to express our gratitude to the reviewers for their positive feedback and recognition of the significance of our findings. The insightful suggestions have helped us improve our manuscript significantly. Below please find the point-by-point response to all the reviewers' comments. In this response letter, the black text is the reviewer's comment, and the blue text is my reply to the above comment.

Point-by-point Responses to Reviewers' Comments:

Reviewer #1 (Remarks to the Author):

The authors have made a commendable effort in addressing all major and minor points that were detailed in the initial review. Portions of the manuscript were revised accordingly and important controls are now included in the biochemical analysis of the circEZH2_005 and hnRNPA1. The initial conclusions made by the authors appear to stand unchanged. The critical and minor points were addressed in a satisfactory manner and the manuscript has been greatly improved by careful revision of writing style and re-working the statistics.

Before publication I ask that the authors consider adding evidence that the intestine subjected to the lengthy ischemic insult followed by reperfusion is not highly necrotic at the starting point their analysis.

Response: We thank the reviewer for the positive comments on our current work. Furthermore, we appreciate the reviewer's concern regarding the potential highly necrosis of the intestine due to prolonged ischemic insult followed by reperfusion. We recognize that the issue should be further validated. Due to the high tolerance of mice, it is often not possible to obtain damaged intestinal tissue samples if the ischemia-reperfusion time is too short. Therefore, in the previously published paper by our research group, we selected an ischemia-reperfusion time that is prone to causing damage for constructing an animal model. Specifically, we established an intestinal I/R animal model with 1 hour of ischemia followed by 2 or 4 hours of reperfusion. (PMID: 36948152, 35241180, 33779497, 36229053). We observed that after clamping the superior mesenteric artery for 60 minutes, about 50% of the mice survived for more than 24 hours. On the other hand, we supplemented the small intestine gross specimens and the corresponding HE staining sections of mice with 1-hour ischemia followed by 2-hour reperfusion. Judging from the gross appearance, the small intestine mainly showed high edema, with only local necrosis (**Figure a**). HE staining also showed similar results (**Figure b**). The sham group exhibited normal intestinal mucosal morphology, while in the I/R group, although part of the mucosa was seriously damaged and the intestinal epithelium was denatured and necrotic with disappearance of the intestinal villi, most areas of the small intestine showed only mild edema. So, we think that the intestine subjected to the lengthy ischemic insult followed by reperfusion is not highly necrotic at the starting point of

our analysis, which is applicable to our research. We thank again for the profound insights provided by the reviewer.

Reviewer #2 (Remarks to the Author):

I believe the author has addressed my question directly. Although there may still be room for discussion and improvement on some aspects, it provides a solid answer that is quite satisfactory in the current research field. This paper should be considered for acceptance.

Response: We are very pleased that the reviewer stratified our revised work. We thank the reviewer for his positive evaluation and meticulous examination of our work.

Reviewer #3 (Remarks to the Author):

Many thanks for revising the manuscript and responding to comments. Amendments are properly replied to and applied within the revised manuscript. Now, I have no further comments and can endorse its publication.

Response: We are very pleased that the reviewer stratified our revised work. We thank the reviewer for positive comments and meticulous examination of our work.

We sincerely thank all reviewers again for your valuable feedback that we have used to improve the quality of our manuscript. Finally, we would like to express our sincere gratitude. Thank you very much for your valuable comments on this work.